# Accelerating invasion potential of disease vector *Aedes aegypti* under climate change

Takuya Iwamura [1✉], Adriana Guzman-Holst [2] & Kris A. Murray [3,4,5✉]

Vector-borne diseases remain a major contributor to the global burden of disease, while climate change is expected to exacerbate their risk. Characterising vector development rate and its spatio-temporal variation under climate change is central to assessing the changing basis of human disease risk. We develop a mechanistic phenology model and apply it to *Aedes aegypti*, an invasive mosquito vector for arboviruses (e.g. dengue, zika and yellow fever). The model predicts the number of life-cycle completions (LCC) for a given location per unit time based on empirically derived biophysical responses to environmental conditions. Results suggest that the world became ~1.5% more suitable per decade for the development of *Ae. aegypti* during 1950–2000, while this trend is predicted to accelerate to 3.2–4.4% per decade by 2050. Invasion fronts in North America and China are projected to accelerate from ~2 to 6 km/yr by 2050. An increase in peak LCC combined with extended periods suitable for mosquito development is simulated to accelerate the vector's global invasion potential.

[1] School of Zoology, Department of Life Sciences, Tel Aviv University, Tel Aviv, Israel. [2] School of Public Health, Imperial College London, London, UK. [3] MRC Centre for Global Infectious Disease Analysis, Department of Infectious Disease Epidemiology, School of Public Health, Imperial College London, London, UK. [4] Grantham Institute - Climate Change and Environment, Imperial College London, London, UK. [5] MRC Unit The Gambia at London School of Hygiene & Tropical Medicine, Atlantic Boulevard, Fajara, PO Box 273, Banjul, The Gambia. ✉email: takuya@tauex.tau.ac.il; kris.murray@imperial.ac.uk

**V**ector-borne diseases continue to be a major contributor to the global burden of disease, annually causing more than 1 billion infections, 1 million deaths, and accounting for around 17% of all lost life, illness and disability globally[1,2]. Insect vectors cannot regulate their internal temperature and are therefore responsive to shifts in climatic conditions over short (e.g. daily weather), medium (e.g. seasons) and long (e.g. El Niño, climate change) time frames[1,3]. Understanding how vectors respond to climatic factors is thus central to characterising the spatio-temporal distributions of vector-borne diseases and anticipating and responding to potential shifts in risk due to climate change.

Although complete congruence cannot be assumed, the incidence and distributions of vector-borne diseases are often conditioned by the distribution and abundance of their primary vectors[1,4,5]. In practice, estimating a species' distribution or abundance is often achieved by estimating some form of environmental suitability[6–10], which broadly defines how favourable a location is for the species with respect to environmental covariates (e.g. temperature, rainfall, habitat). Such an index can take on many forms but is generally derived via either statistical models, which correlate observational records of a species' occurrence with available environmental covariates[11,12], or mechanistic/process-based models, which make use of a species' physiological responses to specific environmental parameters (e.g. temperature, rainfall, humidity, photoperiod; see ref. [13]) typically derived under controlled experimental conditions[14–16]. In both cases, the relationships derived are then projected into geographic space with meteorological (or other environmental) data to assess the changing suitability of landscapes or regions for the species[17,18].

While correlative methods have proved useful for modelling species' distributions on the basis of species' reported occurrence data, when considering potentially invasive disease vectors mechanistic approaches have a number of important advantages in terms of applicability to novel environments[19–21]. In particular, mechanistic models isolate specific biophysical causal pathways that can link an organism's key life-history traits (e.g. development rates, mortality) to its environment, so avoiding reliance on correlations between observed occurrences (which may themselves be inherently biased by e.g. observation effort), environmental covariates and their statistical extrapolation, such as into unoccupied areas or under scenarios of climatic change.

Previous studies have developed mechanistic, temperature-sensitive population dynamics models across multiple life stages for invasive disease vectors, including the Asian tiger mosquito, *Aedes albopictus*[22] and the yellow fever mosquito, *Aedes aegypti*[23]. However, these approaches have rarely been incorporated into distribution estimates, particularly at large spatial and temporal scales to evaluate species' responses to long-term environmental change (but see refs. [16,24]). This is an important research gap given a growing number of primarily statistical studies that have suggested that global climate change may be facilitating the expansion or re-establishment of mosquito vector populations and the diseases they transmit into new or previously occupied regions.

Here, we explore the use of an alternative type of mechanistic model, termed a phenology model[25,26], to examine environmental suitability for the development of the invasive arboviral vector *Aedes aegypti*. Phenology models have been used for predicting invasive pest establishments in agriculture[25,27,28] but they have not, to our knowledge, been applied to invasive human disease vectors. This class of models is characterised by explicitly modelling an organism's physiological development across life-stage transitions according to empirically derived responses to environmental conditions[29]. These responses are used to estimate development rates and critical thresholds, typically from controlled experiments on the species of interest[30]. One advantage of phenology models over other mechanistic models is that they can be used to calculate the number of successful life-cycle completions (LCC) (i.e. the number of generations) per time period, while relying on relatively simple laboratory experiments of temperature-dependent development rates[25].

The objectives of the study are to design, develop and validate a phenology model incorporating the development of each life stage of *Ae. aegypti* and apply it to explore changes in LCC intensity for this vector in response to past and projected climate changes globally. In addition to the annual minimum temperature and precipitation requirements, the phenology modelling framework allows us to use daily climatic inputs to capture the fine scale effects of daily temperature variation on key development rates of mosquitoes at different stages[31,32]. Then, the model makes mechanistic historical projections of LCC intensity over a 100-year period (1950–2050) under RCP 4.5 and RCP 8.5 climate change scenarios, which reflect differences in the degree to which greenhouse gas emissions and consequent climatic changes may be curbed by the middle of this century[33]. Our model predicts increasing and accelerating trends in the LCC of *Ae. aegypti* in the future, as a result of both elevated seasonal peaks in LCC and extended periods of suitable conditions. Invasion frontiers, defined by the climatic suitability at current range margins, in Europe, USA and China are predicted to advance faster in the future into currently unoccupied areas. Determining novel ways to predict and control arthropod vectors and their associated diseases over the coming decades and understanding how deeper emissions cuts could potentially translate into averting increased disease risk are central to supporting adaptation and mitigation strategies to improve global health, socio-economic development and biosecurity strategies in the face of rapid, large scale environmental change[1].

## Results

**Model validation**. We conducted model validation at the two levels—one at the global scale from the occurrence point datasets and another at a local scale using a mosquito abundance dataset. The results showed that our model could broadly reproduce the current known distribution of *Ae. aegypti* globally. Overall, LCC intensity for the midpoint of the time series (2000s as defined by the 2000–2004 average) was highly correlated with the global geographic distribution of *Ae. aegypti* (Fig. 1a), with the vast majority (99.9%) of occurrence records falling in locations with LCC ≥ 1 (Fig. 1b–d). The area under the receiver operating characteristic curve (AUC) was 0.92 at the global scale. Kappa based on the confusion matrix was 0.80 when a LCC > 10 is set as a threshold.

At the country level, AUC was calculated for 12 countries with more than 150 occurrence records (Supplementary Table 1). Even though the model is developed for global-scale study and may lack the ability to consider country-specific situations, in eight of these countries the model predicted national level gradients in occurrence well, with AUCs ranging from 0.63 (moderate; Malaysia) to 0.99 (very good; Taiwan). Higher AUC values were generally obtained in countries with stronger temperature gradients, while in the remaining countries performance was no better than random likely due to being more homogenously suitable and/or poor coverage of observations (e.g. Cuba: 0.47, Indonesia: 0.49) or where lack of available human hosts in remote regions may be limiting (e.g. India: 0.55). Brazil shows low AUC (0.35) likely due to the expanse of the Amazon basin which has a highly suitable climate for *Ae. aegypti* but very low human population densities.

At finer spatial scales, validation tests showed the LCC output had strong ability to provide an index predictive of mosquito

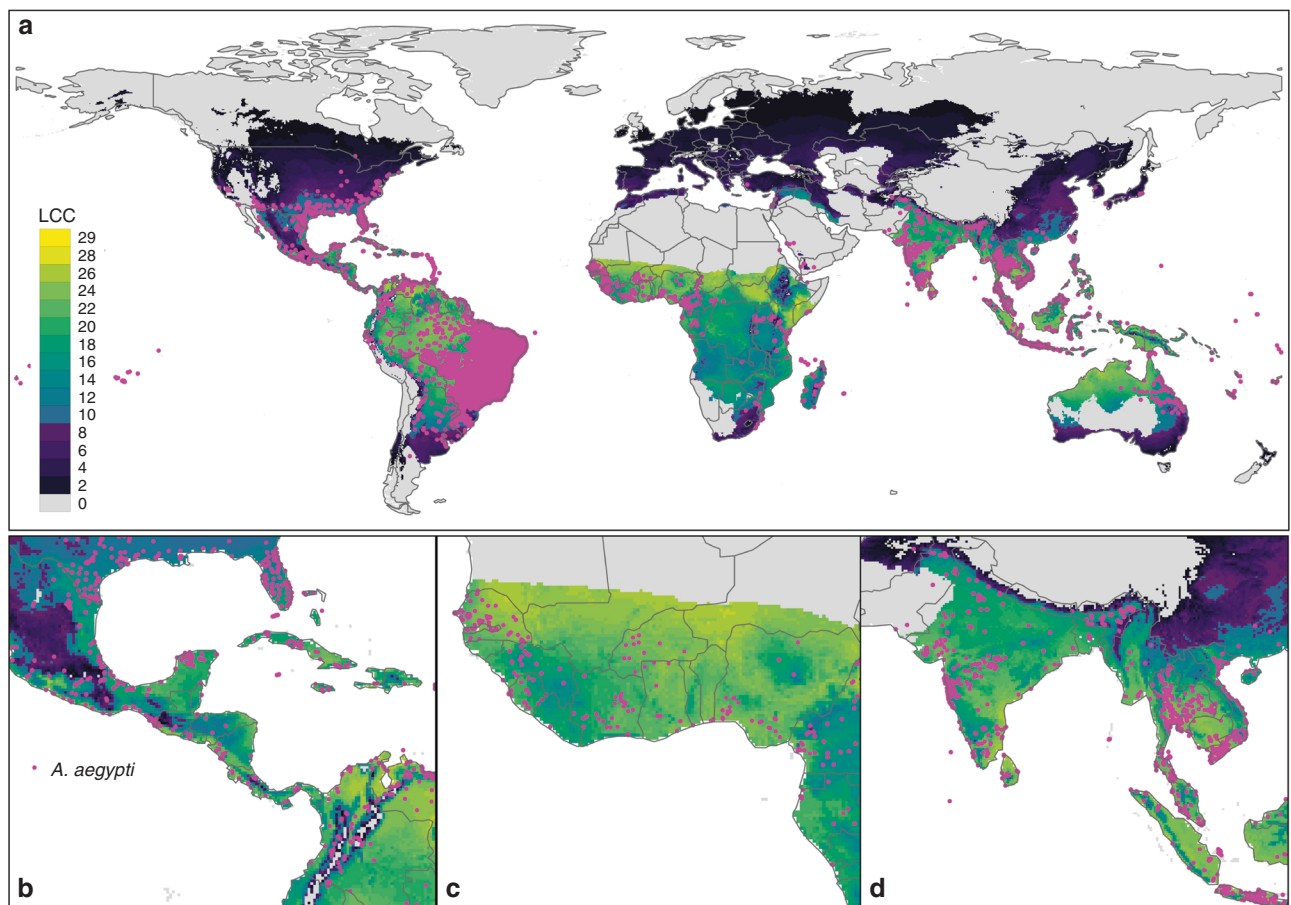

**Fig. 1 Distribution of annual LCC of *Ae. aegypti* with occurrence data overlaid.** Maps indicate the total number of LCC per year at the global scale (**a**), Central America (**b**), West Africa (**c**) and South East Asia (**d**). Colour represents the number of LCC. Areas in which LCC < 10, corresponding to the threshold used in subsequent analysis (i.e. Fig. 4), are shown with a darker palette (indigo-black, note legend). Grey colour represents unsuitable areas for *Ae. aegypti* development. Magenta dots represent presence records of *Ae. aegypti* (refs. [49,84]).

abundance as determined from robust and consistent abundance samples. Model outputs were strongly correlated with the abundance data (Pearson's $r = 0.752$, $r^2 = 0.571$, $p = 0.011$; Supplementary Fig. 1) reported for Mexico in 2011[34,35] (for site details see Supplementary Table 2). These tests suggest that the LCC model output yields process-based information relevant to both occurrence and the population dynamics of this species, which together may underpin establishment risk given successful dispersal into currently uninvaded areas.

**Global trends in *Ae. aegypti* development intensity.** The LCC intensity estimated by the phenology model successfully reproduced the spatial patterns of current *Ae. aegypti* observations as well as local-level abundance (Fig. 1a–d and Supplementary Table 1, see 'Model validation' above for further details). Globally, LCC increased from 7.08 (95% CI across global circulation models (GCMs) 6.96–7.19) per year in the 1950s (1950–1954 average) to 7.62 (7.42–7.82) per year at the turn of the century (2000–2004 average), broadly indicating that the world became ~7.0% (3.1–12.4%) more suitable for the development of this species over this period. Future projections suggest this trend will accelerate, with the average number of generations per year predicted to increase by a further 17.1% (12.4–21.8%) by the 2050s under RCP 4.5 and 24.3% (18.5–30.0%) under RCP 8.5. These changes reflect an acceleration in the increase of global suitability for *Ae. aegypti* development from 1.5% (0.6–2.4%) per decade between the 1950s and 2000s to 3.2% (2.4–4.0%) and 4.4%

(3.5–5.4%) per decade between the 2000s and 2050s under RCPs 4.5 and 8.5, respectively. This amounts to a total predicted change in development intensity of 26.0% (22.1–29.9%) under RCP 4.5 and 33.8% (28.8–38.8%) under RCP 8.5 over the 100-year period considered (1950–2050). Estimations using 10-year averages yielded similar results (see Supplementary Table 3).

Figure 2 illustrates LCC changes with respect to the 2000s average. LCC increased up to 6 LCC per year in tropical areas since the 1950s and a further 6–10 LCC per year is expected by the 2050s in some areas. The greatest increase is predicted under RCP 8.5. Similar increases are estimated using 10-year averages (Supplementary Table 4). The overall suitability, the increase in mean and rates of simulated LCC differ significantly across geographic and climatic regions[36] (Fig. 3, see Supplementary Fig. 2 for the regions). South East Asia, South America and West Central Africa, historically the most suitable regions, are projected to see the greatest increases albeit with higher inter-annual variability, while in Europe, North America and West and Central Asia suitability increases are less pronounced and variable (Fig. 3a). This appears primarily linked to the climatic zones encompassed by these regions, with tropical areas showing particularly high suitability and strong increases in LCC, temperate areas showing marked gains, while arid, polar and boreal climatic zones show low suitability and no or weaker gains (Fig. 3b).

Non-parametric seasonal Kendall trend tests for three monthly time-series datasets (historical 1950–2000, and projected 2000–2050 under RCP 4.5 and RCP 8.5) stratified by latitude

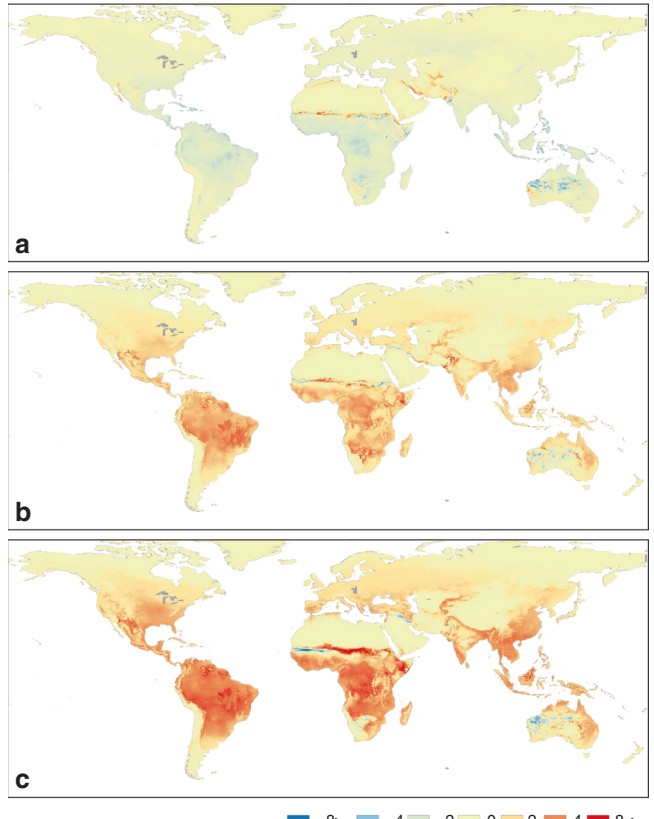

**Fig. 2 Changes of LCC of *Ae. aegypti* between 1950, 2000 and 2050.**
Differences in LCC relative to LCC in 2000s (2000–2004 average).
**a** Comparison with 1950s (1950–1954 average); **b** comparison with 2050s (2050–2054 average) under RCP 4.5; and **c** under RCP 8.5. Decreases in LCC are shown in 'cool' colours (blue and green) and the increases in 'warm' colours (orange and red).

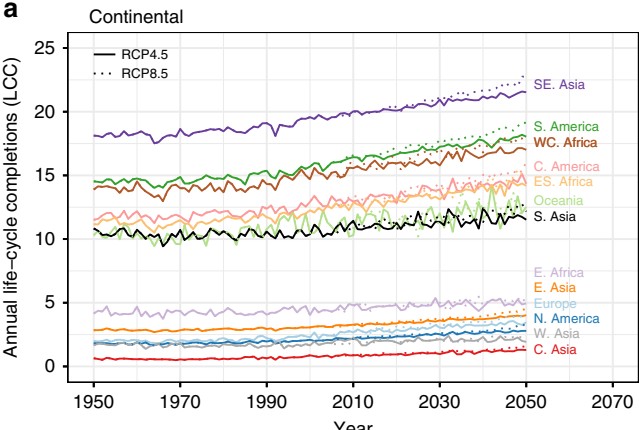

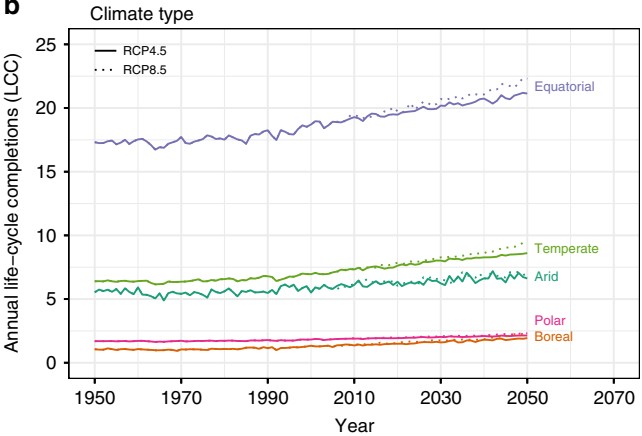

**Fig. 3 Continental and climatic regional averages of annual LCC 1950–2050.** The annual LCC at each location was averaged over continental regions (**a**) and climatic regions (**b**) between 1950 and 2050. Data were calculated for each year with two climatic scenarios—solid lines represent the LCC prediction under RCP 4.5 scenario and dotted lines are under RCP 8.5 scenario. Confidence intervals indicating variability due to the underlying GCMs are not shown here to avoid overplotting and retain clarity of the mean trends—see text for 95% CIs on percentage change statistics over the time series and Fig. 5 for CIs on seasonal trends. Continental regions include Central Asia, East Asia, South Asia, South East Asia, West Asia, Central America, North America, South America, East Africa, East South Africa, West Central Africa, Europe and Oceania. The climatic regions include arid, boreal, equatorial, temperate and polar.

similarly show accelerating future trends relative to the more moderate increases in LCC observed in our results during the historical period. Significant increases in LCC through time since 1950 are observed at all latitudes between 40°N and 40°S, with the slope (rate of change) of these increases growing under future projections, towards the tropics, and under the higher emissions scenario (RCP 8.5) (see Supplementary Fig. 3). For example, the rate of change in LCC per year at 0–10°S in the period 2000–2050, as indicated by the Sen slope estimator, is projected to increase 2.5- and 3.9-fold relative to the historical increase (1950–2000) for RCPs 4.5 and 8.5, respectively (see Supplementary Fig. 4).

**Invasion frontiers.** Contour lines indicating invasion frontiers (≥10 LCC; see "Methods") were used to examine expansion in suitable areas in the three focal regions over multiple periods (Fig. 4). In the USA, the model suggests that the south-eastern states (i.e. Florida, Arizona, Texas) have already seen the advancement of an invasion frontier, as is also supported by observations of *Ae. aegypti* occurrence expanding there. The model confirms relatively slow invasion frontier expansion in China, but predicts more rapid advancement under future climates, including in recent dengue outbreak hotspots (Guangzhou and Guandong provinces)[37]. In Europe, this suitability threshold is patchier, restricted to the southern margins historically, yet clearly increasing suitability in other places (e.g. over the Mediterranean basin) in the future. Continuous stretches of suitability

across Europe are not observed even under RCP 8.5 by 2050 (Fig. 4).

In China, there was only minor change in LCC historically (1950–2000), with the suitability contour line expanding at ~1.58 km year$^{-1}$ (95% CI = 1.41–1.75) km across all the leading edges of the invasion frontier); however, the area predicted to be suitable for population establishment is predicted (2000–2050) to rapidly expand by approximately 5.59 (5.20–5.98) km year$^{-1}$ with a clear north-eastern shift (Fig. 4a). By 2050, much of the populated part of China (south eastern half) is predicted to support ≥10 LCC irrespective of which RCP scenario is considered. In Northern America, a gradual (2.29 (2.12–2.46) km year$^{-1}$) northward expansion of highly suitable areas was predicted historically (1950–2000), while strong future expansion in the USA is also predicted, particularly under RCP 8.5 (5.52 (5.23–5.81) km year$^{-1}$) (Fig. 4b). While *Ae. aegypti* is already present in south-eastern USA, parts of the west are also predicted to become suitable by the 2050s. In Europe, overall suitability

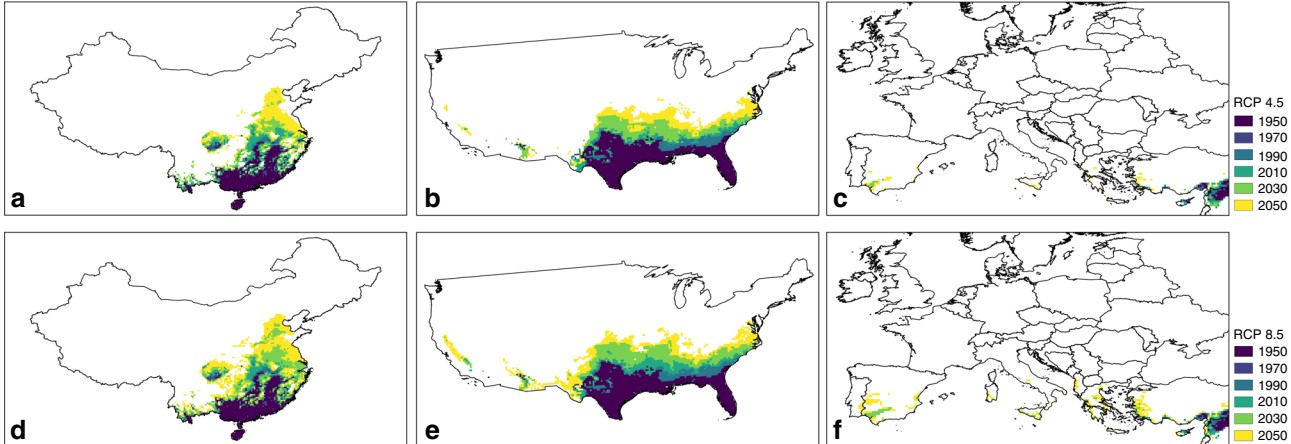

**Fig. 4 Expansion of invasion frontiers of *Ae. aegypti* in China, USA and Europe from 1950–2050 under RCPs 4.5 and 8.5.** Invasion frontiers of *Ae. aegypti* were estimated at decadal intervals for China, USA and Europe based on LCC predictions. The frontier threshold was set at ≥10 LCC based on the frequencies of LCC values extracted to the historical occurrence records of *Ae. aegypti*. The colour scheme represents the frontier contour lines in separate decadal intervals (1950, 1970, 1990, 2010, 2030 and 2050). Shifts in the invasion frontiers are shown for: (**a**) China under RCP 4.5, (**b**) USA under RCP 4.5, (**c**) Europe under RCP 4.5, (**d**) China under RCP 8.5, (**e**) USA under RCP 8.5 and (**f**) Europe under RCP 8.5.

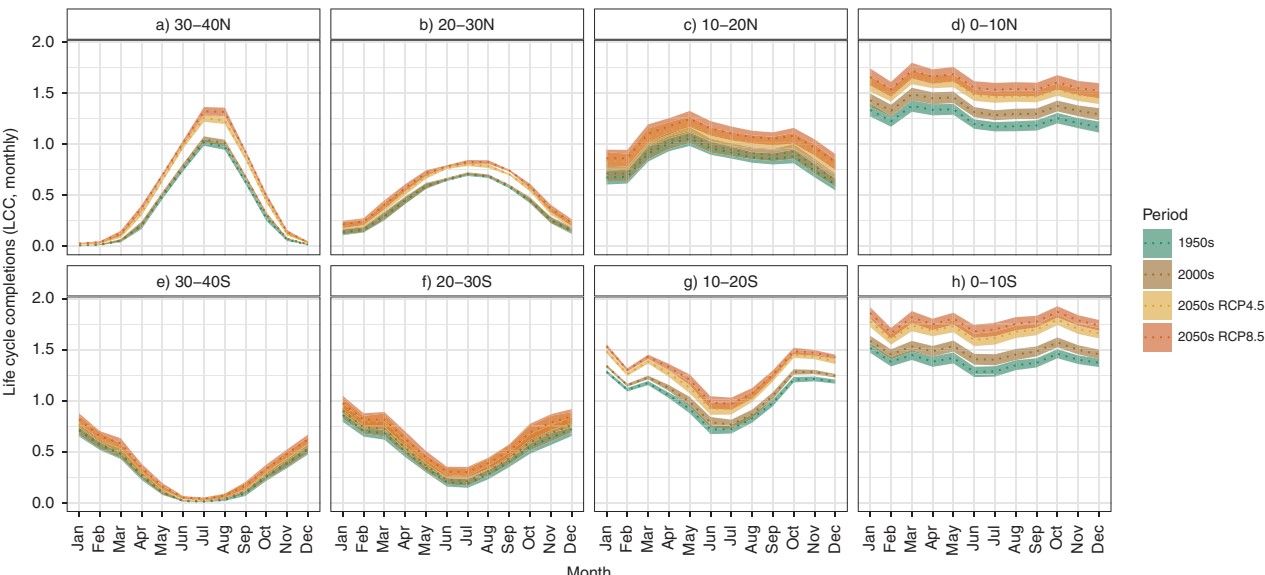

**Fig. 5 Seasonal profiles of LCC based on monthly averages in latitudinal zones.** Monthly average LCC within the latitudinal zones are shown for the years 1950 (1950–54), 2000 (2000–2004) and 2050 (2050–2054) under RCP 4.5 and RCP 8.5 scenarios. The *x* axis represents months from January to December, and the *y* axis is monthly average LCC within each latitudinal zone (**a**)-(**h**) (40°S to 40°N). Colours indicate time period and climate change scenario, where dotted lines indicate means and ribbons around the lines represent 95% CIs to illustrate variability attributable to underlying GCMs.

remains low but southern margins across the Iberian Peninsula, Italy and Greece are predicted to be able to support ≥10 LCC per year from the 2030s, particularly under RCP 8.5 (Fig. 4c).

**Seasonality of average LCC.** Seasonal profiles of LCC for *Ae. aegypti* differed markedly between latitudinal bands (Fig. 5). Overall, seasonality profiles of simulated LCCs suggest increases in development intensity through time during all months and these changes are most pronounced under RCP 8.5. In particular, higher latitudes (20–40°N and S, Fig. 5a, b, e, f) exhibit more significant increases in the duration of the most favourable periods. In addition to surpassing a minimum threshold to support LCC, the expansion in the number of favourable months per year may therefore be a key parameter of interest when considering future mosquito establishment in historically less favourable regions (e.g. outside of the tropics). In contrast, the strongest

increases in peak LCC occur in currently more suitable areas, suggesting these areas may become even more conducive to supporting large mosquito populations.

The lowest and higher latitudes are also predicted to become more favourable during shoulder periods, translating into expanded seasonal profiles in regions outside the tropics. In equatorial regions, while seasonality remains irregular, LCC exhibits a general increase across all months and this is predicted to accelerate by 2050. Importantly, focusing on the comparisons of historical (1950–2000) versus predicted (2000–2050) changes in LCC, historical changes are primarily observable near the equator (0–10°N and S, Fig. 5d, h), while the models indicate future climate changes are predicted to affect all latitudinal bands. The middle latitudes (10–20 and 20–30°N and S) will be most affected with seasonal changes in LCC, with a more significant increase under RCP 8.5 (Fig. 5b, c, f, g).

## Discussion

Climate change is one of the most daunting 21st century global health challenges along with other global environmental and social changes (e.g. land-use changes and accelerated human movement), where expanding distributions and/or increasing abundance of vectors has already begun to reshape certain infectious disease risks[32,38–41]. Here, we sought to translate relatively 'hard' biophysical responses at a very fine temporal resolution (daily) into long-term, global predictions of mosquito LCC to evaluate the response of mosquito development intensity to historical and future climate change. Our model suggests consistent increases in suitability for the LCC of *Ae. aegypti* since 1950, both within the current range (primarily tropics, subtropics) and in currently marginal or uninvaded areas (primarily uninvaded subtropical, currently unsuitable temperate regions), and accelerating increases in suitability to 2050. Increases of LCC in the order of 7% (95% CI 3.1–12.4) globally were predicted by 2000 relative to a 1950s baseline, increasing to 17–24% by 2050 depending on the emissions scenario used. Differences in the LCC under the scenarios RCP 4.5 and RCP 8.5 are expected to diverge even further in future as the effect of cumulative emissions differences become more apparent in the longer term.

The model also sheds light on the idiosyncrasies among regions in the way changing environmental conditions will facilitate vector invasion. Our results predict that invasion frontiers, representing expanding regions that are environmentally suitable for this species, in China and USA are predicted to advance 2.4–3.5 times faster by 2050 (5.2–6.0 km year$^{-1}$) than was estimated through historical projections (1950–2000). Europe is expected to experience isolated areas of sustained suitability for *Ae. aegypti* in Spain, Portugal, Greece and Turkey by 2030. In China, our model predicts expansion of frontiers into the Guangzhou and Guangdong provinces, where dengue outbreaks have been reported recently[37]. These patterns imply that sudden shifts in invasion frontiers should be expected as changing underlying suitability interacts non-linearly with human introduction and dispersal processes (but see ref. [42] for localised speed of the importation of *Ae. aegypti* within established species range).

The seasonal trend analysis further indicates shifting patterns under changing climates in the seasonality of mosquito LCC, whereby both longer periods of favourable conditions and higher intensity in peak LCC are observed in both historical and future projections. Since the duration of favourable periods contributes to cumulative LCC, elongated peak periods could serve an important function in bolstering mosquito abundance in areas with historically stronger seasonality, while increases in peak values could translate to changes in maximum mosquito abundance in favourable periods. These impacts are not uniform but vary considerably across latitudes, with the strongest gains observed in the tropics and subtropics. In contrast to the strongest increases of peak LCC in the equatorial regions, middle latitudes (10–30°N and S) exhibit more significant increases in the duration of the most favourable periods. In addition to surpassing a minimum threshold to support LCC, the expansion in the number of favourable months per year may therefore be a key parameter of interest when considering future mosquito establishment in historically less favourable regions (e.g. outside of the tropics).

Although the trends we report here are clear and validation tests indicate the model performs well in mechanistically replicating observed spatial patterns of *Ae. aegypti* at a global scale, our study has some important limitations. For example, in some regions the model predicts climatic suitability increasing in the future where *Ae. aegypti* has already been observed to be widespread historically (e.g. the Mediterranean and Black Sea region)

or established more recently (resurgence in Black Sea region, parts of the USA including California and Arizona)[43,44]. *Ae. aegypti* was previously introduced into the Americas and the Mediterranean, likely from Africa[45–47], where it vectored outbreaks of yellow fever and dengue (in Europe most recently in Athens during 1927–28[48]) but it had largely retracted from Europe by the 1950s[49–51]. These observations suggest that our model could be too conservative in identifying establishment thresholds in some regions. For example, while at least 1 LCC is broadly predicted across these regions at the beginning of our time series in 1950, theoretically permitting population growth at certain times of the year, it is clear that these regions would not have been reaching a LCC ≥ 10 until significant warming had taken place.

We propose several possible explanations to resolve such inconsistencies. First, areas with low LCC estimates are likely to broadly represent climatically marginal zones for the long-term persistence of *Ae aegypti*. In these areas, extrinsic factors such as management interventions[52] or environmental stochasticity[53] could tip the balance in favour of population extinction more frequently than in highly suitable habitats. This could be the case in Europe, where sporadic observations occur (e.g. Netherlands[54]) and where control measures and cooler winters have previously been hypothesised as causes of 20th century range retractions[55,56]. Second, there are several mechanisms that could result in a mismatch between predicted mosquito responses to climatic variables and what is observed in the field. These include dispersal constraints[57], microclimates (e.g. human infrastructure) and behavioural thermoregulation that would allow species to exploit them[58,59], species interactions (competition, predation)[60], other environmental constraints (e.g. humidity), differences in mosquito responses to climatic constraints (e.g. lineage variation in acclimation ability, tolerance to extremes) or more subtle details of mosquito life-history responses than what is currently captured in our model (e.g. differences in growing degree days (GDD) requirements, development or mortality rates in fluctuating versus mean thermal regimes[61,62]). These can be considered some of the most important areas for future research that could result in further improvements to our model. It is also important to note here that while climate conditions might be suitable for the development and survival of a vector, the conditions for effective disease transmissions may be different. Integrating a disease transmission component into our models or, conversely, integrating our phenology model into existing disease transmission models could help bridge the gap between predicting global change impacts on vectors versus the realized health impacts.

Globally, average temperature increases of over 1 °C have already occurred since the industrial revolution, with even greater warming observed over continental land masses. Our results highlight how such changes have likely already enhanced the potential for *Ae. aegypti*, if introduced, to complete its life cycle in areas with sufficient precipitation and availability of human hosts. Investigations into recent trends in viral disease emergence linked to this vector species, such as the sixfold increase in dengue incidence from 1990 to 2013[63], the establishment and spread of Zika virus in the Americas[64] and recent yellow fever outbreaks in Angola, the Democratic Republic of the Congo and Brazil[65] would be incomplete without closer scrutiny of the role of climate change in bolstering mosquito development or establishment risk alongside other better studied risk factors such as human travel, migration and urbanisation.

Without drastic reduction of greenhouse gas emissions, global temperatures will continue to rise considerably over the coming decades. Our results further suggest that this will continue to favour mosquito development in many regions, both currently inhabited and as yet uninhabited by this invasive species. Such

changes in mosquito population dynamics seem likely to contribute to intensified arboviral disease transmission risks for diseases of considerable public health importance, including dengue, zika, chikungunya and yellow fever. Phenology models provide an additional tool with which to assess the changing basis of arthropod vector population dynamics under climate change.

## Methods

**Study species.** *Aedes aegypti* is the primary vector for several important viruses of global health concern, including dengue, zika, yellow fever and chikungunya. Previous studies have suggested that climate change may have already influenced or will influence the distribution of suitable environments in which *Ae. aegypti* can thrive, resulting in changes in disease risk[50,66]. Beyond its health relevance, *Ae. aegypti* is an ideal species to use for exploring the utility of phenology models for human disease vectors because appropriate data for model parameterisation and validation are broadly available in the literature. This mosquito vector develops from the egg through a number of larval instars before emerging into adults that are only then able to transmit pathogens as the females seek blood meals to support egg development[67].

**Model overview.** Individual life stages and the population dynamics of *Ae. aegypti* are sensitive to environmental conditions, including temperature and rainfall[1,3,68], which can impact the distribution, abundance and invasion potential of the vector. In order to incorporate the environmental sensitivity of development responses of *Ae. aegypti*, we developed a spatially explicit phenology model incorporating each life stage of this mosquito species, following an existing phenology modelling framework[25]. Our model calculates viability for the completion of each development stage (e.g. larvae, pupae, adult) and determines the total number of successful LCC (i.e. generations) possible per time step (e.g. month, year) given gridded climate data input. This output can be considered a development intensity index that theoretically relates to a species' probability of occurrence, abundance and establishment potential (given successful dispersal) in specific locations. Model building and all subsequent statistical analyses were performed in R, version 3.6.1.

**Model structure.** The model incorporates the two main life-cycle periods of *Ae. aegypti*, the aquatic period (immature development) and the aerial period (adult development). The aquatic period is comprised of four stages for growth (eggs, larvae, pupae and adult emergence), while the aerial period is comprised of four adult stages (mating, blood feeding, gestating and oviposition). To simplify the life cycle, the model merges and re-divides development into four sub-stages: (1) egg hatching, (2) immature development (larvae + pupae), (3) blood feeding and (4) oviposition. These four main development stages were formulated into the model based on temperature thresholds and development rates during *Ae. aegypti* development (Supplementary Fig. 5). Only females were considered in the model as they are the egg producing sex. The model, similar to other studies[69], therefore assumes that there are always sufficient males to fertilize females to allow full LCC if environmental conditions allow.

The model comprises two distinct components, GDD (growing degree days) and thresholds, which together determine if and when the development of each life stage (see Rectangles in Supplementary Fig. 5) starts and completes according to prevailing environmental conditions (see diamonds in Supplementary Fig. 5). Thresholds define upper or lower limit conditions that must be met in order for the model to progress or otherwise stop. In contrast, GDD refers to the accumulation of daily mean temperature values over a baseline temperature and below a cut-off temperature beyond which there is no additional benefit[70,71]. GDD calculations include a temperature threshold to mark the point at which development can occur, but thresholds are also applied elsewhere in the model on their own (e.g. heat kill, cold kill, see below). GDD is calculated based on temperature input using the formula

$$\text{GDD} = \sum_i (T_i - T_{thr}),$$
(1)

$T_i = 35$, when above 35,
GDD = 0, when below 0,

where *i* is the number of days at a particular life stage, *T* is the daily average temperature of the *i*th day since the start of the life stage (capped at 35 °C), and thr is the lower temperature threshold of development. On any given day, if $T_i < T_{thr}$ then $T_i - T_{thr}$ is set =0 to reflect that the accumulation of GDD cannot be negative. In components of the model that utilise GDD, if the calculated degree day (from the input climate data) is larger than the established degree day required for development/life-stage completion (derived from the literature), then development is successful and the model moves on to the next life stage. Where this condition is not met, the model cannot progress to complete a full life cycle and this failure is recorded. Conditions derived from the literature to parameterize this component of the model for each life stage are presented below (each stage numbered in Supplementary Fig. 5):

(1) Egg hatching: A temperature-dependent egg hatching time was set considering the lower temperature threshold (baseline temperature) and GDD. Based on estimates from the literature, if the minimum temperature of the day is ≥14.59 °C and GDD since oviposition is 42.4, then the egg hatches[31,32,67]. The hatch time–temperature relationship consists of faster hatching times during warmer temperatures.

(2) Immature development (larvae, pupae): After completion of the egg hatching condition, the larvae must undergo a specific amount of GDD in order to complete the aquatic immature stage and emerge as an adult. We set the baseline temperature as 11.78 °C and 126.38 GDD[67].

(3) Blood feeding: The pre-blood meal period is the time from adult emergence until the first blood meal. This period is temperature sensitive and can be as short as 1 day. It was conditioned as follows: at <20 °C, 4 days; at >20 °C, 2 days; at >26 °C, 1 day, at >35 °C, 2 days (adapted from ref.[67]).

(4) Oviposition: After feeding on the first blood meal, the adult undergoes a temperature-sensitive gonotrophic cycle (GC) to gestate and lay eggs. This period is temperature sensitive and can be as short as 2 days. It was conditioned as follows: at <26 °C, 8 days; at >26 °C, 3 days; at >30 °C, 2 days, at >35 °C, 4 days[52,56].

At the oviposition stage we also imposed a cold-kill condition, considering winter temperature is a key limiting factor for *Ae. aegypti* eggs to persist in the environment[56,72]. To mark the beginning of a period of unsuitably cold temperatures, a condition is imposed on both temperature and duration of the cold period. If the average daily temperature is lower than the set 'cold-kill' temperature threshold for a specific number of days, then the egg dies and cannot proceed with the life cycle. The baseline model was set with the condition of <0 °C for 152 days, based on general consensus from the literature[56,72,73].

In addition to the model stages considered, we imposed a heat-kill condition that applied to all stages outlined above. We set an upper temperature threshold, above which development at any stage (e.g. immature and oviposition) fails to progress. Data from the literature[31,32,74–76] for the upper temperature thresholds for egg hatching and larval development are ≥36 and ≥36.5 °C, respectively. The heat-kill condition for the adult lifespan is >37 °C[31]. Therefore, we set the overall heat-kill condition to any daily temperature above 38 °C for 1 day, applied at any life stage.

In addition to the temperature-dependent parameters, we included a precipitation constraint. After the temperature-sensitive conditions are met, the model imposes a precipitation constraint that removes areas considered too dry to support this species. Based on the Köppen–Geiger climate classification[77] for dry regions and the literature[78,79], the precipitation threshold was set to less than 200 mm of annual rainfall, which contains 99% of observation records. We also examined the model outputs with 900 mm rainfall (95% observation records) as a sensitivity analysis (see Supplementary Table 4), which indicates the general trends are maintained with the higher threshold.

**Sensitivity analyses.** Given that each of our parameters are derived by integrating values from the literature, which contain inherent uncertainties related to, for example, mosquito strain, different methodologies and so on, we conducted sensitivity analyses to explore the effects of using higher or lower values for each parameter (see Supplementary Fig. 6a–c for oviposition, heat-kill, cold-kill analyses, and Supplementary Table 4 for varying the rainfall threshold analysis).

**Climate data.** Gridded climate data were obtained from the NASA Earth Exchange Global Daily Downscaled Projections (NEX-GDDP) dataset[80]. This database provides daily minimum and maximum near-surface air temperature and precipitation from 1950 through 2100 with a spatial resolution of 0.25°, corresponding to about 30 × 30 km grid cells at the equator. Its daily temporal resolution matches the resolution at which mosquitos develop according to prevailing conditions, and for its global coverage, relatively high spatial resolution and bias-corrected climate change projections[81]. NEX-GDDP utilises the Coupled Model Intercomparison Project Phase 5 (CMIP5) GCMs and provides projections for two of four greenhouse gas emissions scenarios corresponding to two levels of radiative forcing (W/m²) by 2100, RCPs 4.5 and 8.5[82,83]. RCP 4.5 represents a middle-of-the-road 'stabilization' scenario by 2100 and corresponding mean global warming of around 2.4 °C, while RCP 8.5 represents a 'business-as-usual' scenario with rising emissions, no stabilization and corresponding warming of around 4.9 °C by 2100[33]. We took an ensemble/consensus approach[32,38,82,83], averaging outputs from four commonly used GCMs for each RCP scenario to capture inter-model variability[32,38]: the Beijing Climate Center Climate System Model, China (BCC-CSM1.1); the National Institute for Environmental Studies Climate Model, Japan (MIROC-ESM-CHEM); the Institut Pierre-Simon Laplace Climate Model, France (IPSL-CM5A-LR) and the National Center for Atmospheric Research's Community Climate System Model, United States (CCSM 4). We chose these GCMs to cover both 'warm' and 'cold' models, which are available from the NEX-GDDP. Equilibrium climate sensitivity (ECS) is at the highest in the MIROC-ESM (ECS 4.7) among all 21 GCMs from CMIP5, while it is among the lowest in BCC1-CSM1.1 (ECS 2.8) and CCSM 4 (ECS 2.9). IPSL-CM5A-LR provides an intermediate (ECS 4.1). An alternative high quality gridded climate dataset, the ECMWF ERA5 hourly climate data, is available and more directly based on observation records; however, at the timing of writing it only covers the period from 1979 to 2019 and use of inconsistent datasets for historical and future climates makes comparisons between these two periods difficult. As such, we present

results from the NASA NEX-GDDP data (1950–2050) but provide sensitivity analyses (long-term trends and spatial averages) to compare the two gridded climate datasets for the period 1980–2005 (see Supplementary Figs. 7 and 8).

**Model validation**. We assumed that LCC is a minimum requirement for establishment and hence occurrence, and that a greater number of LCC should correlate with both higher probability of occurrence and greater abundance. Before applying the model, the ability for the model to predict locations where *Ae. aegypti* can theoretically complete full life cycles was therefore validated using (1) existing *Ae. aegypti* occurrence records and (2) regional-scale abundance data.

The model output (i.e. LCC) was first evaluated against occurrence records of *Ae. aegypti*[84], which includes a global geographic dataset of occurrence records for *Ae. aegypti* derived from both published and unpublished sources, including national entomology surveys and expert communications. This is currently the largest available standardised global dataset for *Ae. aegypti*, with about 40,000 georeferenced observations[84]. The predicted LCC results averaged over the years 2001–2010 were compared with known observation records of *Ae. aegypti* restricted to the same period and summary statistics computed. AUC was examined for the model's ability to discriminate areas of occurrences from areas where it has not been observed based on the observation of other mosquito species as pseudo-absence points, following the same methods as in ref. [49]. For this analysis, AUC, Kappa based on confusion matrix and Pearson correlation between observation and absence were calculated at the global-scale analysis. To calculate the AUC, the life-cycle values were first standardised on a 0–1 scale, where 1 equates the maximum number of life cycles. For Kappa, we set the LCC threshold to consider presence as 10 LCC, which we defined as invasion frontier. In addition to the global-scale validation, we also calculated country-level metrics to see if our global model can be informative at the local scale.

Second, we compared LCC predictions with *Ae. aegypti* abundance data from Lozano-Fuentes et al.[34], who conducted *Ae. aegypti* abundance surveys in villages distributed across a large elevational gradient (0–2000 m) in central Mexico[34,35]. This study was conducted in 2011 and represents the best available case study that we are aware of focusing on *Ae. aegypti* abundance. The surveys spanned a relatively large geographic area (300 × 100 km) and, critically, employed the same sampling methodology across all surveys. We utilised data from all villages in the study, excluding Orizaba and its 'dormitory' city of Rio Blanco[35], leaving surveys from ten villages for analysis (see Supplementary Table 2). For the validation analysis, we extracted the average LCC of our model at the locations for these villages in the same year of the sampling effort (2011) and correlated these values against abundance estimates from the surveys (adult mosquitos). As such our comparison is relative rather than absolute.

**Global spatial and temporal trends in LCC intensity**. Spatially explicit historical and future projections under the two RCP scenarios (4.5, 8.5) over the 100-year period (1950–2050) were conducted to evaluate whether and where environmental suitability for the LCC of *Ae. aegypti* has changed through time due to recent and projected climate change. In addition to presenting the gridded global LCC output maps for comparison with *Ae. aegypti* occurrence records and across select time periods, we averaged LCC over a number of relevant spatial (e.g. continental, climate type, latitudinal bands; Supplementary Fig. 2) and temporal (e.g. yearly, monthly; see below) scales in order to assess overall changes of LCC intensity at these scales through time. Throughout the analysis, we used 5-year averages to define and better assess key time periods: we refer to the 1950s as the 1950–1954 average, the 2000s as the 2000–2004 average and the 2050s as the 2050–2054 average.

**Invasion frontiers**. *Ae. aegypti* has expanded its global range in recent decades, with changes in environmental suitability at invasion fronts likely facilitating this spread in some instances. We thus investigated how changes in predicted LCC due to climate change at three focal invasion fronts (USA, China, Europe) could contribute to expanding ranges. To do this, we defined an invasion frontier contour as a contour line representing the LCC value below which 2.5% of *Ae. aegypti* occurrence records globally occurred, representing uncommon but demonstrated establishment at the lower end of the LCC distribution. Globally, this 2.5% contour line corresponds with the areas ≤10 LCC per year, so this was set as the invasion frontier threshold. We then tracked this invasion frontier contour through time to illustrate which new areas could become suitable for future establishment and by when. Invasion speed was estimated by average minimum distance between sampled points on a leading edge of an invasion frontier contour of a target year (e.g. 2050) to comparative invasion frontier contour of a previous period (e.g. 2000). The number of sampled points ranged from 300–700, depending on the shape and length of an invasion frontier contour.

**Seasonality of LCC intensity**. In many parts of the world, mosquito population dynamics are strongly seasonal, contributing to temporal variations in risk for vector-borne diseases[85]. In particular, mosquito populations may be influenced by the seasonal effects in both peak development and the overall duration of periods in which development is enhanced. To evaluate potential changes in the seasonality of development intensity for *Ae. aegypti* under climate change, we estimated LCC for

each grid cell monthly for each year in the analysis. We summarised these results in latitudinal categories (0–10, 10–20, 20–30 and 30–40°, N and S) to facilitate interpretation. We also used seasonal Kendall trend tests and Sen slope estimation implemented with the *EnvStats* package in R to evaluate and compare long-term trends across periods and latitudinal bands accounting for seasonality.

## Data availability

The daily temperature and precipitation data that support the findings of this study are available in NEX-GDDP (https://dataserver.nccs.nasa.gov/thredds/catalog/bypass/NEX-GDDP/catalog.html).

## Code availability

All of our code including estimating and analysing LCC are freely available at the GitHub repository (https://github.com/takuyaiwamura/vector_lcc). A demo package including a small scale test dataset is available at the Zenodo repository (https://doi.org/10.5281/zenodo.3701852).

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

## Acknowledgements
T.I. received the start-up fund from Tel Aviv University for this study. A.G.-H. received scholarship from Imperial College London.

## Author contributions
T.I. and K.A.M. designed the study. T.I. developed the phenology model. A.G.-H. collected physiological data. T.I. and A.G.-H. conducted the simulations. K.A.M. conducted statistical analyses. T.I., A.G.-H and K.A.M. wrote the paper.

## Competing interests
The authors declare no competing interests.
