## [Peer Review File · Nature Communications]

Reviewers' comments:

Reviewer #1 (Remarks to the Author):

Review of "Accelerating invasion potential of disease vector *Aedes aegypti* under climate change"

Manuscript number: NATCOM-19-16916

Authors: Iwamura et al., 2019

Recommendation: Major revisions

General:

This study aims to model the impact of climate change on the distribution and activity of the yellow fever mosquito, *Ae. aegypti*, at global scale. The authors employ a very interesting modelling approach e.g. they apply a phenology model, commonly used to model the life cycle of plants and pests in agriculture, to model the life cycle of the mosquito (each stage of the life cycle depends on degree day models to derive the number of life-cycle completions e.g. LCC). The authors show that climate change will increase suitability for *Ae. aegypti*, with a clear acceleration in suitability shown by 2050. They also show an increase during peak seasons and a potential lengthening of the LCC season in future, in particular for temperate regions.

Overall the paper is well written and there are lots of interesting ideas in there. The use of a phenology model (derived from agriculture), detailed LCC changes per climatic regions, and the investigation of changes in LCC seasonality are very positive points / interesting ideas. However, there are several major (mostly methodological) flaws that need to be addressed before the manuscript can be considered for publication in Nature communications (see my detailed comments below).

Major points:

1] Use long term averages to estimate climate change signals

The authors use a single time point (single year basically) to estimate changes in time. A single year can reflect interannual variability e.g. a particular year can be colder or warmer than average, and this signal can be related to natural climate variability or other external factors (like a volcanic eruption that can cool down the atmosphere for example). The authors should use a 10y average minimum instead (20y averages are commonly employed to assess long term climate change). If you select only one time point (one year) this can lead to very weird results. One example, on Figure 5 (top left panel): simulated LCC in 1950 is larger than LCC in 2000, which is very counter-intuitive given the observed increase in temperature between 1950 and 2000... The authors should use 10y average minimum instead (1950-1959 for example – this can be referred as to 1950s in the text). Similar comment apply to future projections. The 2000-2050 time average is a bit too long as well (median in 2025) as the emission scenarios (RCP4.5 and RCP8.5) won't differ much over this period. The authors could use 2040-2049 (2040s) or 2030-2049 instead.

2] Future scenarios too conservative and more information required for historical background *Ae. aegypti* used to plague southern Europe during the early part of the 20th century. *Ae. aegypti* used to be present in southern France, Italy, Spain, Portugal, Bosnia and Macedonia in the early 1900s (Christophers, 1960; Reiter, 2000). The largest early dengue epidemic occurred in Athens in 1927-28 (Louis, 2012). The historical context for *Ae. aegypti* should be discussed further in the text. The future projections are somehow too conservative I think. In Europe, hotspots are shown over southern Spain, southern Italy, southern Greece, and Turkey for 2050 (RCP8.5 scenario – Fig 4 so for LCC > 10). Hotspots seem to be in the right locations, but temperature conditions in the early 1900s were already suitable for *Ae. aegypti* over southern Europe. The "recent context" e.g. the recently reported presence of *Ae. aegypti* over the eastern coasts of the Black sea and eastern Turkey [<https://ecdc.europa.eu/en/publications-data/aedes-aegypti-current-known-distribution-january-2019>] should be discussed as well. Another example: California and Arizona in the US appears as suitable (Fig 4) in 2050 (RCP8.5). *Ae. aegypti* is already well established over California and Arizona (see Hahn et al., 2016 for up to date surveillance data for the US). The historical background and recent papers based on surveillance & field entomological data should be discussed and caveats of the model outputs discussed further in discussion.

3] Use observed historical gridded climate data to estimate LCC instead of GCM data

The authors employ GCM data to derive historical LCC estimates. The NASA-NEX GCM data has been calibrated with respect to observed climate conditions (both historical and scenario streams). However, the authors should use observed climate data to derive historical estimates of LCC to complement the historical maps derived from the GCM historical simulations. The authors can use ERAINTERIM or ERA5 daily gridded temperature data to achieve such a task.

4] Selection of 5GCMs in the NASA-NEX GDDP ensemble

The authors use a subset of 4 GCMs e.g. BCC-CSM1.1, MIROC-ESM-CHEM, IPSL-CM5A-LR and CCSM4 (out of 21 GCMs available from the NASA-NEX GDDP archive <https://cds.nccs.nasa.gov/nex-gddp/>). They claim that they employed "four commonly used GCMs" and cite two studies, one is using Worldclim data and the other is using CRU data (U. East Anglia). What is the metric defined to subset these 4 climate models out of 21? Please clarify. Ideally, the authors should select the warmest, coldest and median GCM in terms of temperature changes.

Minor points:

L34: "Insect vectors are climate-sensitive" – ok but you could be more precise e.g. mention that arthropods do not regulate their internal temperature, so air temperature is important

L44 : "via either correlative/statistical models" – remove "correlative" as they should be included in "statistical models".

L47: "(e.g. temperature)" – ok but sophisticated models can also incorporate the effect of rainfall and sometimes humidity and other factors – so add more parameters in there

L52: "reported occurrence data" – sounds better

L55: "including the Asian tiger mosquito, *Ae. albopictus*, and the yellow Fever mosquito, *Ae. aegypti*..."

L58: "acknowledgment" – "evidence" instead?

L59: "only a few models..."

L60: "and fewer still influences complete mosquito life-cycles..." Ok this is a selling argument for your study but there are lots of papers with dynamical models for *Aedes* mosquitoes... Perhaps precise "at global scale" – as compartmental models are mostly used at local / regional scale

L77-79: Precise which scenarios briefly e.g. RCP4.5 and RCP8.5 – people are more or less used to the jargon now

L93: "average 2046-2050" – see major point 1

Figure 1: Add occurrence points on global map (perhaps white crosses - Fig 1a).

Figure 2: Discrete bins on Fig 2 are confusing (-6 then -2 then 0; 2; 6 and 10) and difficult to read. Bins should be continuous and centred (-10 to 10 with step = 2 namely -10 till -8 a blue color ... -2 till 2 a white color – 2 till 4 a yellow color ... 8 to 10 red color)

Figure 3: The large interannual variability in simulated LCC for Oceania looks suspicious. Most curves are relatively smooth, while there are large spikes shown for Oceania. This might be related to masking issues and/or spatial averages over islands. Please double check this large variability is not related to artefacts related to spatial averaging.

Figure 3 caption: "E. Africa" – typo

Figure 3 – another comment – as you focus on 2000-2050 the RCP scenarios are relatively similar (they start diverging around 2040 roughly) – this is why you have small differences between RCP4.5 and RCP8.54 on Fig 3b (and you somehow conserve this relationship for $LCC=f(temp)$)

L132: "Overall, only 2.8%..."

Figure 4: You select 10LCC as a threshold to discriminate "invasion frontiers". So I think you should highlight a clear break for the 10LCC bin on Figure 1 (change colour bar use blue scale below 10 – yellow scale below 20 and red scale below 30).

L164: General comment – add degree symbol for spatial coordinates "10-30°N" etc

Figure 5: Add sub labels a), b), c), d) e), f), g) and h) on Figure 5. This way you can refer to a particular panel plot when you discuss results in the text. For example, when you discuss LCC seasonality changes for 30-40°N – you can refer to "Fig 5a" etc

L196: "was poorer likely due..." – not poorer – the exact semantic should be "no better than random" – AUC close to or below 0.5 denotes that the model is no better than random

Fig S3: X-axis label should be "LCC predictions" – Y-axis label should be "Observation [add units]".

Units for observation seem to be abundance per year? Please clarify

L208-210: Climate change is important, I agree. However you should mention globalization and movement of goods and persons as an important risk factor. Climate might become more suitable in pixel X – but if the vector is not introduced then nothing happens

L237: "... the potential for *Ae. aegypti*, if introduced, ..."

References:

S.R. Christophers. 1960. *The Yellow Fever Mosquito—Its Life History, Bionomics and Structure*. By Sir S. Rickard Christophers. Pp. xii + 739. (Cambridge: At the University Press, 1960.)

[http://www.dpi.inpe.br/geocxnets/wiki/lib/exe/fetch.php?media=wiki:christophers_1960.pdf]

Micah B. Hahn, Rebecca J. Eisen, Lars Eisen, Karen A. Boegler, Chester G. Moore, Janet McAllister, Harry M. Savage, John-Paul Mutebi, Reported Distribution of *Aedes (Stegomyia) aegypti* and *Aedes (Stegomyia) albopictus* in the United States, 1995-2016 (Diptera: Culicidae), *Journal of Medical Entomology*, Volume 53, Issue 5, September 2016, Pages 1169–1175=

Louis C. 2012. Daily Newspaper View of Dengue Fever Epidemics, Athens, Greece, 1927-1931. *Emerging Infectious Diseases* 18(1) 78-82.

Reiter, P. 2010. Yellow fever and dengue: a threat to Europe? *Euro Surveill.*15: 19509.

Reviewer #2 (Remarks to the Author):

This manuscript by Iwamura et al. details a temperature driven mosquito life cycle model that is applied globally to attempt to infer patterns about the current and future distribution of arboviral diseases. While this study does have some novel aspects such as assessing change 1950-2000, I felt overall it only made very minor advances to what is now a well-developed body of literature. I also felt there were several major flaws in the modelling approach and validation that make the results very difficult to interpret.

Major comments:

I struggled to see what this added over previous work. Temperature-driven models of *Ae. aegypti* life cycle stages have been around since the late 1990s (e.g. Patz et al. *Environmental Health Perspectives* 1998) with various abstractions of vectorial capacity since to simplify various components of transmission (e.g. Moredecai et al. *PLOS NTDS* 2017, Brady et al. *Parasites and Vectors* 2014). Each of these are arguably more developed, more detailed and better validated than what is presented here.

The developed model uses very specific thresholds for developmental stages. This does not adequately propagate the (large) uncertainty around these thresholds. Many of the stages discussed are also likely to have distributional rather than threshold behaviours and as a result are not well described by a single parameter.

Given the model is purely a function of temperature (the precipitation threshold is essentially arbitrary and not suitable given the highly dynamic interaction between precipitation and mosquito habitat), the authors should really consider whether it is appropriate to assess the current and future distribution of dengue with this model. There are clearly a large variety of other factors (climatic and non-climatic) that affect current and future distribution and without including them it is difficult to see how these results will have any policy relevance.

The model only uses air temperature data, but simulates dynamics of mosquito aquatic stages. Indoor water temperature and its dynamics are likely to be substantially different from average (outdoor) air temperature estimates.

As shown in previous work e.g. Lambrechts et al. PNAS 2011, diurnal variations in temperature are important for predicting suitability, especially in marginal habitats, daily mean temperature is insufficient.

The model validation is not particularly rigorous. Random generation of pseudo absences (ratio not reported) with no radius does not accurately account for the reporting biases in the data nor does it account for spatial sorting bias – leading to overinflated AUCs. Even then the model does very poorly in some very important areas for global spread e.g. Brazil AUC = 0.35). Surely areas at the fringes are where predictive skill is most relevant? Even in the best case scenario with the Mexico abundance data where temperature is the clear most likely candidate for a driver of the distribution, correlation is only marginally significant

The Discussion has no limitations section despite the study having many limitations.

Minor comments:

Timing of blood feeding data comes from a Laboratory experiment and doesn't consider delays imposed by host-seeking.

Cold kill of OC seems over generous given adults typically can't survive temperatures below (10-15C)

Line 370- averaging GCMs does not account for inter-model variability + why this subset of GCMs?

2.5 vs 2.8 % as invasion threshold- different values stated in the results and methods sections.

Line 241-247- attributing causality between climate change and one off arbovirus emergence events seems impractical. I would suggest that either the authors suggest a statistically robust experiment to test this, or remove it.

Responses to reviews (reviews received July 5th, 2019)

Dear Dr Iwamura,

I have received the referees' reports on your manuscript entitled "Accelerating invasion potential of disease vector *Aedes aegypti* under climate change". You will see from their comments copied below that, while they find your work of considerable potential interest, they raise quite substantial concerns that must be addressed. In light of these comments, we cannot accept the manuscript for publication, but would be interested in considering a substantially revised version that deals with those concerns.

I hope you will find the referees' comments useful. Please bear in mind that my colleagues and I will be reluctant to approach the referees again in the absence of major revisions. However, do not hesitate to contact me if there are specific requests that you believe are technically impossible or unlikely to yield a meaningful outcome. If the revision process takes significantly longer than three months, we will be happy to reconsider your manuscript at a later date, as long as nothing similar has been accepted for publication at Nature Communications or published elsewhere in the meantime.

When resubmitting your paper, please highlight all changes in the manuscript text file. We also ask that you ensure that your manuscript complies with our editorial policies. Specifically, please ensure that the following requirements are met, and any relevant checklists are completed or updated and uploaded as a Related Manuscript file type with the revised article:

Reviewers' comments:

Reviewer #1 (Remarks to the Author):

*Review of "Accelerating invasion potential of disease vector *Aedes aegypti* under climate change"*

Manuscript number: NATCOM-19-16916

Authors: Iwamura et al., 2019

Recommendation: Major revisions

General:

*This study aims to model the impact of climate change on the distribution and activity of the yellow fever mosquito, *Ae. aegypti*, at global scale. The authors employ a very interesting modelling approach e.g. they apply a phenology model, commonly used to model the life cycle of plants and pests in agriculture, to model the life cycle of the mosquito (each stage of the life cycle depends on degree day models to derive the number of life-cycle completions e.g. LCC). The authors show that climate change will increase suitability for *Ae. aegypti*, with a clear acceleration in suitability shown by 2050. They also show an increase during peak seasons and a potential lengthening of the LCC season in future, in particular for temperate regions.*

Overall the paper is well written and there are lots of interesting ideas in there. The use of a phenology

model (derived from agriculture), detailed LCC changes per climatic regions, and the investigation of changes in LCC seasonality are very positive points / interesting ideas. However, there are several major (mostly methodological) flaws that need to be addressed before the manuscript can be considered for publication in Nature communications (see my detailed comments below).

Author response (in bold):

Thank you for these positive and very constructive comments. As detailed below we have responded and conducted additional analyses to address all of the concerns raised. This includes re-running the model for the 'historical' range using the new dataset suggested (ERA5) for the years available (1980-2019). We have also clarified a number of key points or rewritten sections for clarity. Overall, we are confident that we have addressed all of the points raised by the reviewer, and this makes our manuscript much stronger.

Major points:

1] Use long term averages to estimate climate change signals

The authors use a single time point (single year basically) to estimate changes in time. A single year can reflect interannual variability e.g. a particular year can be colder or warmer than average, and this signal can be related to natural climate variability or other external factors (like a volcanic eruption that can cool down the atmosphere for example). The authors should use a 10y average minimum instead (20y averages are commonly employed to assess long term climate change). If you select only one time point (one year) this can lead to very weird results. One example, on Figure 5 (top left panel): simulated LCC in 1950 is larger than LCC in 2000, which is very counter-intuitive given the observed increase in temperature between 1950 and 2000... The authors should use 10y average minimum instead (1950-1959 for example – this can be referred as to 1950s in the text). Similar comment apply to future projections. The 2000-2050 time average is a bit too long as well (median in 2025) as the emission scenarios (RCP4.5 and RCP8.5) won't differ much over this period. The authors could use 2040-2049 (2040s) or 2030-2049 instead.

Response:

This is a good point and we apologize for the confusion. In fact, we did use 5-year averages as specified in the Methods section for the comparison of select time points for the global analyses but this was not reflected in the label or caption in the figures. We have clarified this, and in addition now conducted 10-year averages as the reviewer suggested (we deemed 20-year average is too long for our study). We chose to keep the 5-year averages in the main text, but present results for the 10-yr window in the Supplementary Information (see the Table S3). As can be seen, the differences between the two windows are negligible (e.g. 26.0 and 27.4 as the increased % between 1950s to 2050s under RCP 4.5 in the 5- and 10-yr windows respectively).

Regarding Figure 5, many thanks for catching this - we found an error in this calculation and indeed the number is coming from a single year. Therefore, we took 5 year-average for this graph, and have now corrected, which brings the figure in line with expectation. Now the graph shows expected patterns with 1950 and 2000 showing similar patterns in the higher latitudinal bands (30-40 degrees).

2] Future scenarios too conservative and more information required for historical background
Ae. aegypti used to plague southern Europe during the early part of the 20th century. *Ae. aegypti* used to be present in southern France, Italy, Spain, Portugal, Bosnia and Macedonia in the early 1900s (Christophers, 1960; Reiter, 2000). The largest early dengue epidemic occurred in Athens in 1927-28 (Louis, 2012). The historical context for *Ae. aegypti* should be discussed further in the text. The future projections are somehow too conservative I think. In Europe, hotspots are shown over southern Spain, southern Italy, southern Greece, and Turkey for 2050 (RCP8.5 scenario – Fig 4 so for LCC > 10). Hotspots seem to be in the right locations, but temperature conditions in the early 1900s were already suitable for *Ae. aegypti* over southern Europe. The “recent context” e.g. the recently reported presence of *Ae. aegypti* over the eastern coasts of the Black sea and eastern Turkey [<https://ecdc.europa.eu/en/publications-data/aedes-aegypti-current-known-distribution-january-2019>] should be discussed as well. Another example: California and Arizona in the US appears as suitable (Fig 4) in 2050 (RCP8.5). *Ae. aegypti* is already well established over California and Arizona (see Hahn et al., 2016 for up to date surveillance data for the US). The historical background and recent papers based on surveillance & field entomological data should be discussed and caveats of the model outputs discussed further in discussion.

Response:

Thank you for these insights. We have now added an address of these historical observations in relation to our results in the Discussion. We now write (L269-296):

“...in some regions the model predicts climatic suitability increasing in the future where *A. aegypti* has already been observed to be widespread historically (e.g., the Mediterranean and Black Sea region) or established more recently (resurgence in Black Sea region, parts of the USA including California and Arizona)^{37,38}. *A. aegypti* was previously introduced into the Americas and the Mediterranean, likely from Africa³⁹⁻⁴¹, where it vectored outbreaks of yellow fever and dengue (in Europe most recently in Athens during 1927-2842) but it had largely retracted from Europe by the 1950s⁴³⁻⁴⁵. These observations suggest that our model could be too conservative in identifying establishment thresholds in some regions. For example, while at least 1 life-cycle completion is broadly predicted across these regions at the beginning of our time series in 1950, theoretically permitting population growth at certain times of the year, it is clear that these regions would not have been reaching an LCC ≥ 10 until significant warming had taken place.

We propose several possible explanations to resolve such inconsistencies. First, areas with low LCC estimates are likely to broadly represent climatically marginal zones for the long-term persistence of *A. aegypti*. In these areas, extrinsic factors such as management interventions⁴⁶ or environmental stochasticity⁴⁷ could tip the balance in favour of population extinction more frequently than in highly suitable habitats. This could be the case in Europe, where sporadic observations occur (e.g., Netherlands⁴⁸) and where control measures and cooler winters have previously been hypothesised as causes of 20th century range retractions^{49,50}. Second, there are several mechanisms that could result in a mismatch between predicted mosquito responses to climatic variables and what is observed in the field. These include dispersal constraints⁵¹, microclimates (e.g., human infrastructure) and

behavioural thermoregulation that would allow species to exploit them^{52,53}, species interactions (competition, predation)⁵⁴, other environmental constraints (e.g., humidity), differences in mosquito responses to climatic constraints (e.g., lineage variation in acclimation ability, tolerance to extremes), or more subtle details of mosquito life-history responses than what is currently captured in our model (e.g., differences in GDD requirements, development or mortality rates in fluctuating vs mean thermal regimes^{55,56}). These can be considered some of the most important areas for future research that could result in further improvements to our model.”

3] Use observed historical gridded climate data to estimate LCC instead of GCM data

The authors employ GCM data to derive historical LCC estimates. The NASA-NEX GCM data has been calibrated with respect to observed climate conditions (both historical and scenario streams). However, the authors should use observed climate data to derive historical estimates of LCC to complement the historical maps derived from the GCM historical simulations. The authors can use ERAINTERIM or ERA5 daily gridded temperature data to achieve such a task.

Great point. We agree that we could use observed climate data for direct comparison with GCM reconstructions for historical estimates of LCC. ERA 5 (ERA Interim is an obsolete version of ERA 5) is a great suggestion but it is not fit for our purposes as it only extends back to 1979 and has not been calibrated to link to future projection scenarios in GCMs in common use (or vice versa). Nevertheless, we have now included additional analysis using the ERA5 dataset for the years 1980-2005 for a comparison with our results using NEX GDDP data over the same period and included this in the Supplementary Information (post-2005 data from NEX GDDP utilizes the IPCC scenarios, RCP 4.5 and 8.5 and so were not compared). The results show that there are differences in the results (as expected), but the observed trends through time and space are similar (Fig. S6 and S7). We have noted some key apparent differences (primarily tropical areas) in the SI. Thus, we see both pros and cons for the use of these two datasets. Ultimately, however, ERA5 cannot serve as a replacement for our purposes of long-term change assessments requiring 100 years of data with calibrated backcasts and forecasts available for this period (as the reviewer mentioned, NASA NEX is calibrated with the Global Meteorological Forcing Dataset (Sheffield et al. 2006).

4] Selection of 5GCMs in the NASA-NEX GDDP ensemble

The authors use a subset of 4 GCMs e.g. BCC-CSM1.1, MIROC-ESM-CHEM, IPSL-CM5A-LR and CCSM4 (out of 21 GCMs available from the NASA-NEX GDDP archive <https://cds.nccs.nasa.gov/nex-gddp/>). They claim that they employed “four commonly used GCMs” and cite two studies, one is using Worldclim data and the other is using CRU data (U. East Anglia). What is the metric defined to subset these 4 climate models out of 21? Please clarify. Ideally, the authors should select the warmest, coldest and median GCM in terms of temperature changes.

We completely agree and have revised the text to clarify. We did choose the 4 GCMs to cover both ‘warm’ and ‘cold’ models. Equilibrium climate sensitivity (the temperature increase from sustained doubling of the concentration of CO₂ in the Earth's atmosphere) is at the highest in the MIROC-ESM (4.7) among all 21 GCMs from CMIP5, while it is amongst the lowest in BCC1-CSM1.1 (2.8) and CCSM 4 (2.9). IPSL-CM5A-LR provides somewhere in the middle (4.1). This description is included in the main text (L.438-443).

Minor points:

L34: "Insect vectors are climate-sensitive" – ok but you could be more precise e.g. mention that arthropods do not regulate their internal temperature, so air temperature is important

Thank you for pointing this out. We revised our description (L34-36) as "Insect vectors cannot regulate their internal-temperature and are therefore responsive to shifts in climatic conditions over short (e.g., daily weather), medium (e.g., seasons) and long (e.g., El Niño, climate change) time frames (Hales et al. 2002; Campbell-Lendrum et al. 2015)."

L44 : "via either correlative/statistical models" – remove "correlative" as they should be included in "statistical models".

We followed the suggestion and removed the correlative.

L47: "(e.g. temperature)" – ok but sophisticated models can also incorporate the effect of rainfall and sometimes humidity and other factors – so add more parameters in there

Agreed. We incorporate more factors and this section now reads (L48) "(e.g. temperature, rainfall, humidity; see ¹³)."

Citation 13 is (Tjaden et al. 2018) Mosquito-Borne Diseases: Advances in Modelling Climate-Change Impacts. Trends in Parasitology.

L52: "reported occurrence data" – sounds better

Fixed.

L55: "including the Asian tiger mosquito, Ae. albopictus, and the yellow Fever mosquito, Ae. aegypti..."

Fixed.

L58: "acknowledgment" – "evidence" instead?

Followed the suggestion and changed to 'evidence'.

L59: "only a few models..."

Fixed.

L60: "and fewer still influences complete mosquito life-cycles..." Ok this is a selling argument for your study but there are lots of papers with dynamical models for Aedes mosquitoes... Perhaps precise "at global scale" – as compartmental models are mostly used at local / regional scale

Agreed and added 'at global scale' as suggested. Now it reads (L52-67): "While correlative methods have proved useful for modelling species' distributions on the basis of species' reported occurrence data, when considering potentially invasive disease vectors mechanistic approaches have a number of important advantages in terms of applicability to novel environments¹⁸⁻²⁰. In particular, mechanistic models isolate specific biophysical causal pathways that can link an organism's key life-history traits (e.g., development rates, mortality) to its environment, so avoiding reliance on correlations between observed occurrences (which may themselves be inherently biased by e.g., observation effort), environmental covariates and their statistical extrapolation, such as into unoccupied areas or under scenarios of climatic change.

Previous studies have developed mechanistic, temperature-sensitive population dynamics models across multiple life stages for invasive disease vectors, including the Asian tiger mosquito, *Aedes albopictus*²¹ and the yellow fever mosquito, *A. aegypti*²². However, these approaches have rarely been incorporated into distribution estimates, particularly at large spatial and temporal scales to evaluate species' responses to long-term environmental change (but see²³). This is an important research gap given a growing number of primarily correlative studies that have suggested that global climate change may be facilitating the expansion or re-establishment of mosquito vector populations and the diseases they transmit into new or previously occupied regions."

L77-79: Precise which scenarios briefly e.g. RCP4.5 and RCP8.5 – people are more or less used to the jargon now

Fixed.

L93: "average 2046-2050" – see major point 1

We revised the text for clarity and added analyses with 10-yr averages to the SI. Please see response to the major point 1.

Figure 1: Add occurrence points on global map (perhaps white crosses - Fig 1a).

Thanks for the suggestion. Occurrence points were not originally included in Fig 1a due to the points obscuring the results layer underneath. We have now added them, however, but retained the magenta color as in the other panels as this provided the strongest contrast with the results layer and other map features (e.g., white background).

Figure 2: Discrete bins on Fig 2 are confusing (-6 then -2 then 0; 2; 6 and 10) and difficult to read. Bins should be continuous and centred (-10 to 10 with step = 2 namely -10 till -8 a blue color ... -2 till 2 a white color – 2 till 4 a yellow color ... 8 to 10 red color)

Thank you for suggestion. We agree and changed the color scheme as suggested.

Figure 3: The large interannual variability in simulated LCC for Oceania looks suspicious. Most curves are

relatively smooth, while there are large spikes shown for Oceania. This might be related to masking issues and/or spatial averages over islands. Please double check this large variability is not related to artefacts related to spatial averaging.

Thank you for pointing this out. We confirm this is the result of the spatial operation we used. We have corrected the method and the Oceania region now shows similar but lower magnitude variability (See Fig. 3).

Figure 3 caption: "E. Africa" – typo

Fixed.

Figure 3 – another comment – as you focus on 2000-2050 the RCP scenarios are relatively similar (they start diverging around 2040 roughly) – this is why you have small differences between RCP4.5 and RCP8.54 on Fig 3b (and you somehow conserve this relationship for $LCC=f(temp)$)

Thank you for your insights. We had also noted this in the Discussion (L246-248): "Differences in the LCC under the scenarios RCP4.5 and 8.5 are expected to diverge even further in future as the effect of cumulative emissions differences become more apparent in the longer term."

L132: "Overall, only 2.8%..."

Fixed.

Figure 4: You select 10LCC as a threshold to discriminate "invasion frontiers". So I think you should highlight a clear break for the 10LCC bin on Figure 1 (change colour bar use blue scale below 10 – yellow scale below 20 and red scale below 30).

Thank you for the suggestion. We have now adjusted the colour scheme to match, making use of two separate but visually compatible scales on either side of the threshold (on the upper side we use the original palette, on the lower side we use a darker palette to demarcate the threshold at <10LCC).

L164: General comment – add degree symbol for spatial coordinates "10-30°N" etc

Done.

Figure 5: Add sub labels a), b), c), d) e), f), g) and h) on Figure 5. This way you can refer to a particular panel plot when you discuss results in the text. For example, when you discuss LCC seasonality changes for 30-40°N – you can refer to "Fig 5a" etc

Done.

L196: "was poorer likely due..." – not poorer – the exact semantic should be "no better than random" – AUC close to or below 0.5 denotes that the model is no better than random

We have now revised this section to address this comment (L223): “Higher AUC values were generally obtained in countries with stronger temperature gradients, while in the remaining countries performance was no better than random...”

Fig S3: X-axis label should be “LCC predictions” – Y-axis label should be “Observation [add units]”. Units for observation seem to be abundance per year? Please clarify

Fixed. It is adult abundance during the survey season (summer of 2011). The graph and text have been revised.

L208-210: Climate change is important, I agree. However you should mention globalization and movement of goods and persons as an important risk factor. Climate might become more suitable in pixel X – but if the vector is not introduced then nothing happens

Agree. We have revised the text, which now reads (L236-229): “Climate change is one of the most daunting 21st century global health challenges along with other global environmental and social changes (e.g. land-use changes and accelerated human movement), where expanding distributions and/or increasing abundance of vectors has already begun to reshape certain infectious disease risks^{31,32,34–36}.”

L237: “... the potential for Ae. aegypti, if introduced, ...”

Fixed.

References:

S.R. Christophers. 1960. The Yellow Fever Mosquito—Its Life History, Bionomics and Structure. By Sir S. Rickard Christophers. Pp. xii + 739. (Cambridge: At the University Press, 1960.)

[http://www.dpi.inpe.br/geocxnets/wiki/lib/exe/fetch.php?media=wiki:christophers_1960.pdf]

Micah B. Hahn, Rebecca J. Eisen, Lars Eisen, Karen A. Boegler, Chester G. Moore, Janet McAllister, Harry M. Savage, John-Paul Mutebi, Reported Distribution of Aedes (Stegomyia) aegypti and Aedes (Stegomyia) albopictus in the United States, 1995-2016 (Diptera: Culicidae), Journal of Medical Entomology, Volume 53, Issue 5, September 2016, Pages 1169–1175=

Louis C. 2012. Daily Newspaper View of Dengue Fever Epidemics, Athens, Greece, 1927-1931. Emerging Infectious Diseases 18(1) 78-82.

Reiter, P. 2010. Yellow fever and dengue: a threat to Europe? Euro Surveill.15: 19509.

Reviewer #2 (Remarks to the Author):

This manuscript by Iwamura et al. details a temperature driven mosquito life cycle model that is applied globally to attempt to infer patterns about the current and future distribution of arboviral diseases.

While this study does have some novel aspects such as assessing change 1950-2000, I felt overall it only made very minor advances to what is now a well-developed body of literature. I also felt there were several major flaws in the modelling approach and validation that make the results very difficult to interpret.

Many thanks for this summary and recognition that our manuscript contains novel aspects. We are, however, sorry that we did not convey the full range of novel aspects more effectively. We have redoubled our efforts in this revision to clearly describe these and other novel aspects of our study, including those also highlighted in Review 1. These include:

- **Application of a mathematical development stage model for *A. aegypti* to a global scale spatial analysis**
- **Use of a daily climate dataset to capture very fine scale biological processes of mosquito development, while focusing on representing intermediate-term (seasonal) and long-term term trends**
- **Results showing climate change is causing an *acceleration* in environmental suitability for the development of *A. aegypti* by 2050**
- **Results showing how changing seasonality contributes to changing long term trends in development potential (i.e., increase in the number of life cycles during peak seasons and longer peak season in future, in particular for temperate regions).**

We have also responded to all the specific points raised and included some additional analyses to address the concerns (see also responses to Review 1). In particular, we have now improved the validation method by using more refined background points as used in the definitive work of Kramer et al., on *A. aegypti* distribution modeling and we have included additional routinely used validation metrics to better communicate the predictive performance of our model.

We also stress that our model has not been developed for dengue specifically, as stated or implied in several comments in Review 2, and as such we contend that it is unreasonable to benchmark or directly compare our results against existing dengue studies. Instead, our results describe the spatio-temporal patterns of the mosquito *A. aegypti*'s ability to complete full life-cycles (life cycle completions, LCC), making it highly relevant to wider range of important arboviruses despite not being a transmission-oriented, disease specific model.

More broadly, our approach is novel for human disease vectors and as such could serve as an important inspiration for other researchers to explore and further develop this simple yet powerful class of models to open up perennially needed new avenues of research relevant to infectious disease epidemiology, ecology, and management.

Major comments:

*I struggled to see what this added over previous work. Temperature-driven models of *Ae. aegypti* life cycle stages have been around since the late 1990s (e.g. Patz et al. *Environmental Health Perspectives* 1998) with various abstractions of vectorial capacity since to simplify various components of*

transmission (e.g. Moredecai et al. PLOS NTDS 2017, Brady et al. Parasites and Vectors 2014). Each of these are arguably more developed, more detailed and better validated than what is presented here.

This is a fair comment and we accept responsibility for the novelty of our contribution not being made more explicit. As mentioned above we have concentrated in this major revision in making the novelty of our work crystal clear. In addition, as stated above, Review 2 appears to be confusing our study with previous studies dealing with disease transmission. Although some of these models do contain mosquito life-history traits as part of their formulation, some of which are temperature driven for certain parameters as mentioned, they generally do not allow assessment of the kind that we are making here (changes in LCC through time linked to long term climate trends both historical and projected, based on daily input data).

The developed model uses very specific thresholds for developmental stages. This does not adequately propagate the (large) uncertainty around these thresholds. Many of the stages discussed are also likely to have distributional rather than threshold behaviours and as a result are not well described by a single parameter.

The reviewer raises an important point regarding the mechanisms used in our model, and we agree there is likely to be some spatial variation in the parameters incorporated. We have added a reference to this in the limitations section in the Discussion.

The comment also highlights that our original explanation was not adequate for readers to understand our model correctly – it is not simply a threshold-based model as the reviewer has interpreted. It uses both growing degree days (GDD), which captures accumulated temperature through time based on daily input data, as well as other thresholds (e.g., Tmax thresholds for different life stages). Due to the mathematical nature of the GDD, the timing of moving from one stage to next is non-linear and dependent on local conditions. We have now added additional detail to Fig S1 to clarify this aspect.

Combining daily temperature, which fluctuates every day, the resulting timing of moving from one stage to the next is ‘distributional’, rather than start/stop behavior often observed through simple threshold-based models. We would also highlight that predictive performance of our model has been extensively tested and found to be very good globally and in many cases more locally, refuting the suggestion that our model is not generally adequate due to threshold uncertainty.

Given the model is purely a function of temperature (the precipitation threshold is essentially arbitrary and not suitable given the highly dynamic interaction between precipitation and mosquito habitat), the authors should really consider whether it is appropriate to assess the current and future distribution of dengue with this model. There are clearly a large variety of other factors (climatic and non-climatic) that affect current and future distribution and without including them it is difficult to see how these results will have any policy relevance.

Again, we reiterate this is not a dengue specific nor a disease transmission modeling paper and direct comparisons with existing dengue studies is not a constructive approach here. Please see also comments above.

The model is based on reconstructing mosquito biology with temperature-dependence integrated into model parameters, so it is incorrect to regard our model as “purely a function of temperature”. Many population dynamics and very many epidemiological models do not contain any environmental inputs at all.

It is not clear why the precipitation threshold is considered inappropriate considering this is, at our spatial scale for a global model, well supported by data on defining the distributional limits of this species. When compared to global precipitation patterns and known *A. aegypti* observations, our original threshold (200 mm) captures 99.5% of all observations globally, which we felt was a conservative approach to avoid predictions in the most unsuitable regions. We have nevertheless included an extra analysis in this revision using a less conservative threshold of 900mm (which removes around 5% of all observations globally). Results of the new analysis are consistent with the previous results, with the key findings regarding accelerating trends in mosquito development threshold still being observed (see SI). Global scale validation indicates that the two settings are similarly accurate based on Kraemer et al.’s presence and pseudo-absence datasets (AUC ~ 0.9, Kappa ~ 0.8). Please see the corresponding section below for the detailed explanation on the validation process.

We do of course agree with the more general point that other factors are also important for further defining the distribution and population dynamics of vector species, and that such factors are likely to become increasingly important as spatial scales become very local. Indeed, ecological studies are frequently conducted within the powerful theoretical framework of ecological niche theory, which broadly defines potential and realized species distributions and population dynamics as being constrained by dispersal and any number of potentially limiting factors and their interactions, both biotic (e.g., competition, predation) and abiotic (e.g., environmental factors), and in many cases this may include social or other human factors as well. We have now added references to these issues in the Discussion.

Nevertheless, we suggest that in this case our model is ‘fit for purpose’ in focusing primarily on the temperature and precipitation axes of this species’ ecological niche – we are explicitly favoring a mechanistic model for its generally superior qualities in representing relatively hard bio-physical relationships to make projections beyond the current range of this invasive species and under future climate conditions, situations in which extrapolating statistical models is often considered inappropriate. This comes with a trade-off in our ability to incorporate a more diverse range of covariates that could reasonably be hypothesized to affect the distribution and dynamics of vector species (as is common and more straightforward when using statistical models, including ENMs). This point is also recognized by Steiner et al., (2013) in their review of mechanistic vectorial capacity models for vector-borne diseases, who noted that despite decades of research such models remain fairly similar in many respects to the original Ross-McDonald formulation and still only rarely attempt to include environmental covariates and complete mosquito life-cycles. We would also again restate that we have included extensive validation metrics that broadly indicate the model performs well, and of course so that readers can evaluate the predictive performance of the model for themselves.

The model only uses air temperature data, but simulates dynamics of mosquito aquatic stages. Indoor

water temperature and its dynamics are likely to be substantially different from average (outdoor) air temperature estimates.

We combined the water availability and accumulated temperature GDD. For this study, outdoor air temperature was used as a proxy for water temperature for mosquito egg laying and larval development sites (typically small pools of water in outdoor settings rather than indoors). Because these aquatic stages occur very near to the surface of the water, the air temperature is indeed a reasonable proxy for the environment of mosquito development (Ritchie et al. 2014);(Christophers 1960);(Hopp & Foley 2001). We have also added some detail in the Discussion (limitations section) regarding the more general issue of the potential mismatch between model parameterization and the conditions actually experienced by mosquitoes in the field.

As shown in previous work e.g. Lambrechts et al. PNAS 2011, diurnal variations in temperature are important for predicting suitability, especially in marginal habitats, daily mean temperature is insufficient.

The citation the reviewer mentioned (Lambrechts et al. PNAS 2011 “Impact of daily temperature fluctuations on dengue virus transmission by *Aedes aegypti*”) showed that the disease transmission of dengue can be influenced by diurnal variation. Although we again restate that our paper focuses on mosquito development and not disease transmission specifically, the paper does show that “Mosquitoes lived longer ... under moderate temperature fluctuations”. More relevant to our use of GDD, a more recent study by Carrington et al. (2013, Plos One) showed that “observed degree-day estimates for mosquito development under fluctuating temperature profiles depart significantly (around 10–20%) from that predicted by constant temperatures of the same mean.” (we cited both of these studies in our paper).

Although we did not make this explicit, we decided not to incorporate diurnal temp. range (DTR) impacts on GDD calculations into our phenology model given the additional complexity this introduces to the model, the lack of appropriate data to effectively capture the effect of DTR on the specific parameters in our model, and the lack of consensus around the predictability of such effects across life-stages and species (Kutcherov & Lopatina 2018) and through time (Verheyen & Stoks 2019). Instead, we calculated GDD using the mean of Tmin and Tmax values for each day. We have now included text to reflect this issue in the text and acknowledge DTR effects in the limitations section (L. 267-296, in particular L. 291-299): “These include dispersal constraints⁵¹, microclimates (e.g., human infrastructure) and behavioural thermoregulation that would allow species to exploit them^{52,53}, species interactions (competition, predation)⁵⁴, other environmental constraints (e.g., humidity), differences in mosquito responses to climatic constraints (e.g., lineage variation in acclimation ability, tolerance to extremes), or more subtle details of mosquito life-history responses than what is currently captured in our model (e.g., differences in GDD requirements, development or mortality rates in fluctuating vs mean thermal regimes^{55,56}). These can be considered some of the most important areas for future research that could result in further improvements to our model.”

The model validation is not particularly rigorous. Random generation of pseudo absences (ratio not reported) with no radius does not accurately account for the reporting biases in the data nor does it account for spatial sorting bias – leading to overinflated AUCs. Even then the model does very poorly in some very important areas for global spread e.g. Brazil AUC = 0.35). Surely areas at the fringes are where

predictive skill is most relevant? Even in the best case scenario with the Mexico abundance data where temperature is the clear most likely candidate for a driver of the distribution, correlation is only marginally significant

We have now improved the validation to reflect these suggestions in two ways – 1) the inclusion of weighted pseudo-absence data in place of random pseudo-absences and 2) the use of an additional validation metric, Kappa.

For 1), we now conduct AUC validation using the same pseudo-absence dataset Kraemer et al. has used to control to the extent possible the effects of observation/reporting bias, which is based on the observation records of other *Aedes* species. This, as Kraemer et al. also state, is only applicable for global scale analysis. In contrast to the suggestion that our original validation result was ‘inflated’ due to the use of random pseudo-absences, we find that both AUC (global 0.924) and Kappa (global 0.82) increase further when using the background data from Kraemer et al. We have revised these statistics in the text to match.

Since countries with very high *A. aegypti* observations also tend to have much lower number of observation records of other *Aedes* species, we cannot use these data as pseudo-absence for country level validation as AUC is unreasonably inflated. The Kappa metric is similarly unsuitable at this scale as it requires country-specific LCC thresholding (thus we cannot apply the standardized validation). Thus we kept our original AUC scores as validation.

We would, however, emphasize here that, due to the nature of global analyses, country level validation is of relatively limited value and we have reported these statistics primarily as a convenience for readers. We acknowledge several times in the ms that our outputs are likely to miss country-specific idiosyncrasies while our results show that this global model nevertheless does a reasonable job in predicting country-level presence for most countries with only a few exceptions (Table S2). For each country with a low AUC, we have hypothesised why the low AUC may be observed. Results for Brazil, for example, are difficult to validate with observation records due to the Amazon basin – it may be suitable for mosquitos, but the human population is extremely low and observation effort is highly limited.

With respect to the validation of abundance prediction on Mexico survey data, the correlation is not distributional (i.e, absence/presence) as implied in Review 2 – it is the correlation with abundance and the number of life cycle completions (LCC) that the model predicted. As such, we feel this is a highly informative validation step that strengthens our ability to interpret our results, suggesting that our results are indeed indicative of mechanisms related to population dynamics as measured independently through capture surveys at fairly local scales. We have used a standard statistical approach in assessing the correlation and a significance test with $p < 0.05$ to evaluate this result. We acknowledge the ongoing debate around the interpretation of p values and significance testing (e.g., <https://www.nature.com/articles/d41586-019-00857-9>) but disagree that the test and p value (Pearson’s $r = 0.752$, $r^2 = 0.571$, $p = 0.011$) in this case should be simply dismissed as ‘marginal’ support for a what appears to be an interesting and relevant biological effect that should help catalyse further work on this topic in future.

The Discussion has no limitations section despite the study having many limitations.

We have now incorporated a consolidated limitations section in the revision (L. 270-299): “Although the trends we report here are clear and validation tests indicate the model performs well in mechanistically replicating observed spatial patterns of *A. aegypti* at a global scale, our study has some important limitations. For example, in some regions the model predicts climatic suitability increasing in the future where *A. aegypti* has already been observed to be widespread historically (e.g., the Mediterranean and Black Sea region) or established more recently (resurgence in Black Sea region, parts of the USA including California and Arizona)^{37,38}. *A. aegypti* was previously introduced into the Americas and the Mediterranean, likely from Africa^{39–41}, where it vectored outbreaks of yellow fever and dengue (in Europe most recently in Athens during 1927-2842) but it had largely retracted from Europe by the 1950s^{43–45}. These observations suggest that our model could be too conservative in identifying establishment thresholds in some regions. For example, while at least 1 life-cycle completion is broadly predicted across these regions at the beginning of our time series in 1950, theoretically permitting population growth at certain times of the year, it is clear that these regions would not have been reaching an LCC ≥ 10 until significant warming had taken place.

We propose several possible explanations to resolve such inconsistencies. First, areas with low LCC estimates are likely to broadly represent climatically marginal zones for the long-term persistence of *A. aegypti*. In these areas, extrinsic factors such as management interventions⁴⁶ or environmental stochasticity⁴⁷ could tip the balance in favour of population extinction more frequently than in highly suitable habitats. This could be the case in Europe, where sporadic observations occur (e.g., Netherlands⁴⁸) and where control measures and cooler winters have previously been hypothesised as causes of 20th century range retractions^{49,50}. Second, there are several mechanisms that could result in a mismatch between predicted mosquito responses to climatic variables and what is observed in the field. These include dispersal constraints⁵¹, microclimates (e.g., human infrastructure) and behavioural thermoregulation that would allow species to exploit them^{52,53}, species interactions (competition, predation)⁵⁴, other environmental constraints (e.g., humidity), differences in mosquito responses to climatic constraints (e.g., lineage variation in acclimation ability, tolerance to extremes), or more subtle details of mosquito life-history responses than what is currently captured in our model (e.g., differences in GDD requirements, development or mortality rates in fluctuating vs mean thermal regimes^{55,56}). These can be considered some of the most important areas for future research that could result in further improvements to our model.”

Minor comments:

Timing of blood feeding data comes from a Laboratory experiment and doesn't consider delays imposed by host-seeking.

Most studies regarding timings of blood feedings are laboratory based, yet we adapted the data from Carrington et al. (2013) based on the fact that the feeding can be delayed/take longer depending on

the temperature. So for example, <20C it can take the host 4 days to feed and at >35C it can take 2 days. By doing this, we are indirectly showing there might be some delay in blood feeding. This is described in the Methods section.

Cold kill of OC seems over generous given adults typically can't survive temperatures below (10-15C)

Cold kill applies to egg stage from the literature on non-diapausing eggs, cold hardiness, and supercooling points for *Aedes* eggs. We considered data from different regions (for generalizability) and time-periods. Please refer to the following references: Thomas et al 2012, Mogi et al 1995, Hanson et al 1995, Christophers et al 1960.

Line 370- averaging GCMs does not account for inter-model variability + why this subset of GCMs?

We calculated model outputs (LCC) for each GCM. We present both the mean and the variation between model outputs (as upper and lower 95% confidence intervals) so our results do illustrate inter-model variability.

We chose these GCMs to cover the wide range of GCM predictions based on equilibrium climate sensitivity (ECS; the temperature increase from sustained doubling of the concentration of CO₂ in Earth's atmosphere). ECS is at the highest in the MIROC-ESM (4.7) among all 21 GCMs from CMIP5, while it is among the lowest in BCC1-CSM1.1 (2.8) and CCSM 4 (2.9). IPSL-CM5A-LR provides somewhere in the middle (4.1). Please refer to Response 4 of Review 1 for more details.

2.5 vs 2.8 % as invasion threshold- different values stated in the results and methods sections.

Corrected to 2.5%.

Line 241-247- attributing causality between climate change and one off arbovirus emergence events seems impractical. I would suggest that either the authors suggest a statistically robust experiment to test this, or remove it.

We did not attribute such causality between climate change and one off arbovirus emergence events – we suggested that the role of climate via its effect on vector development and establishment should be considered alongside other factors (from L306): “Investigations into recent trends in viral disease emergence linked to this vector species, such as the six-fold increase in dengue incidence from 1990 to 2013⁵⁷, the establishment and spread of Zika virus in the Americas⁵⁸ and recent yellow fever outbreaks in Angola, the Democratic Republic of the Congo and Brazil⁵⁹ would be incomplete without closer scrutiny of the role of climate change in bolstering mosquito development or establishment risk alongside other better studied risk factors such as human travel, migration and urbanization”.

REFERENCES

Campbell-Lendrum D, Manga L, Bagayoko M, Sommerfeld J. 2015. Climate change and vector-borne diseases: what are the implications for public health research and policy? Philosophical

Transactions of the Royal Society B: Biological Sciences **370**:20130552. The Royal Society.

Christophers R. 1960. *Aedes Aegypti* (L.) the yellow fever mosquito: Its life history, bionomics and structure. Page Cambridge At The Universit Press.

Hales S, De Wet N, Maindonald J, Woodward A. 2002. Potential effect of population and climate changes on global distribution of dengue fever: an empirical model. *The Lancet* **360**:830–834. Elsevier.

Hopp MJ, Foley JA. 2001. Global-scale relationships between climate and the dengue fever vector, *AEDES AEGYPTI*. *Climatic Change*.

Kutcherov D, Lopatina EB. 2018. Growth and development under variable temperatures revisited: the importance of thermal thresholds. *Zoology and Ecology*.

Ritchie SA, Buhagiar TS, Townsend M, Hoffmann A, van den Hurk AF, McMahon JL, Eiras AE. 2014. Field Validation of the Gravid *Aedes* Trap (GAT) for Collection of *Aedes aegypti* (Diptera: Culicidae) . *Journal of Medical Entomology*.

Sheffield J, Goteti G, Wood EF. 2006. Development of a 50-year high-resolution global dataset of meteorological forcings for land surface modeling. *Journal of Climate*.

Tjaden NB, Caminade C, Beierkuhnlein C, Thomas SM. 2018. Mosquito-Borne Diseases: Advances in Modelling Climate-Change Impacts.

Verheyen J, Stoks R. 2019. Temperature variation makes an ectotherm more sensitive to global warming unless thermal evolution occurs. *Journal of Animal Ecology*.

Reviewers' comments:

Reviewer #1 (Remarks to the Author):

Review of "Accelerating invasion potential of disease vector *Aedes aegypti* under climate change"

Manuscript number: NATCOM-19-16916A

Authors: Iwamura et al., 2019

Recommendation: Major revisions - still

The paper greatly improved since the last round of revisions. However, I still spotted important points that deserve clarifications. The manuscript plan needs to be restructured a bit as well (validation section should come first in results); and there are minor points to address before the paper can be accepted for publication in Nature communications.

Major points:

A - Model validation should come first in the results section. The authors also need to state that they carried out a two level validation – one at global scale (using points in and out of their suitability surface) – one more detailed for Mexico (this needs to be spelled out in the text)

B - Items in Supplementary Materials should be cited in a chronological order in the text (starting with Table S1 – Table S2 etc - Fig S1 - S2 etc) so the authors will have to reshuffle the items in Supp Material accordingly.

C – Something important I missed in the former round of revisions (sorry about that) – the authors state that Invasion fronts in North America and China are projected to accelerate from ~2 to 6km/yr by 2050. The recent work by Kraemer et al 2019 [Fig 1 at <https://www.nature.com/articles/s41564-019-0376-y>] shows that historical invasion fronts (derived from observed vector presence data from 1990 to 2010) were more likely in the 60 km/year range for the US and 100km/year for Europe. Thus, there is a large discrepancy between simulated invasion fronts for the future based on this study (about 20ish km in 10 years) and recently observed invasion fronts (about 60 or 100 km in one year for temperate regions – and this is consistent with the observed spread of *Ae. albopictus* from Italy to the north of France in roughly 10ish years). This might be related to the fact that the mosquito already invaded its potential niche during the recent context and it will slowly spread in future due to climate change (and this finding is consistent with the Kraemer paper). Other recent studies on a similar topic should also be discussed further in discussion (Monaghan work- <https://journals.plos.org/ploscompbiol/article?id=10.1371/journal.pcbi.1007369> - Kraemer work on invasion front <https://www.nature.com/articles/s41564-019-0376-y> etc). The authors need to discuss this point further in discussion (perhaps use relative changes and state that the historical invasion fronts estimated from their model are quite small with respect to rates derived from observations for Europe and the USA).

Minor comments:

L19-20: "change in human disease risk" - singular sounds better

L20-21: "we develop a mechanistic phenology model..." - you can state the novelty of your study there: "we develop a mechanistic phenology model derived from agriculture and applied it to..."

L21: "an invasive vector of impactful arboviruses"

L26: "while this trend will accelerate..." - when you mention the future and risk simulations, everything is hypothetical (so avoid deterministic statement like this) so reword to "while this trend is simulated to..."

L28: "during peak seasons" - "during warm seasons"

Last sentence of the abstract: "An increase in LCC combined with a lengthening of the transmission season is simulated to accelerate ..."

L41: "...are often linked to or partially predicted by the distribution..." - "are often conditioned by"

Note this is not a necessary true conditions - theoretically, climate conditions might be suitable for a particular vector (allowing development and survival), but temperature conditions might still be too low to enable the insect vector to become infectious - in other words they can survive but

won't be able to transmit disease in location X or Y. As an example, some endemic UK mosquito species can transmit exotic pathogens (like Zika virus for example) in a lab setting at high temperatures (work in progress in our team), but they won't be able to transmit the pathogen in the field (so temperature are still too cold to allow mosquitoes to become infectious). I could have made that comment earlier but this needs to be discussed in the manuscript.

L50-51: "to assess the static or changing suitability of landscapes or regions for the species" - "to assess the stability (or stationarity) of landscapes or regions for the species"

L70 - (Linnaeus, 1762) - ok but remove it - no need to cite the father of taxonomy in here I would say

L71 - "for predicting invasive pest establishments..." - "for predicting invasive pest establishments affecting agriculture production" - mention agriculture in that sentence and you can say these methods have not been extensively applied to medical entomology (to state the novelty of your study further).

L84: "...key development rates in mosquitoes..." - "development rates of mosquitoes at different stages"

L85: RCP8.5 (remove space between RCP and scenario ID - double check the whole doc as I spotted this several time).

"RCP4.5 and RCP 8.5 climate change scenarios, which reflect differences in the degree to which greenhouse gas emissions and consequent climatic changes may be curbed by the middle of this century". Ref 32 refers to a VBD modelling paper - not the official RCPs- authors should cite the standard Moss et al 2010 <https://www.nature.com/articles/nature08823> or Van Vuren et al 2011 paper <https://link.springer.com/article/10.1007/s10584-011-0148-z>

Fig 1 caption - "to the threshold used in Fig 4" - prefer "to the threshold used in subsequent analysis"

Fig 3 – typo in legend - "equatorial" should be "equatorial"

Photoperiod is important for *Ae. albopictus* and it might be a limiting factor at high latitudes (mention this somewhere in discussion)

L160 - Generic comment do not say: [see Discussion] in the text - discuss your results then mention important points missing in discussion please

Fig 5 caption - when referring to geographical domain use standard degrees north / south system eg [40S to +40°N for example] - or the reader might be confused with degrees Celsius (for temperature)

L219: "from something to something".

L229: "reported for Mexico by" - you need to state that you have extra validation analysis for Mexico somewhere in Methods.

L346: "eggs, larvae, pupae and emergence" - "eggs, larvae, pupae and adults" – I am unsure I think some arrows are missing on Fig S1 (connecting eggs to life cycle)

Table S1: The authors should highlight AUC above 0.7 (in bold – good scores) – AUC below or equal to 0.5 threshold are not better than random – so use a color or star code to discriminate good and poor AUC values at country scale. The low AUC values for Brazil are not surprising (as models tends to forecast the whole country as suitable – and surveillance data is mostly available for the populated districts / eastern regions), and I assume that collecting mosquitoes in the Amazon is quite complicated.

Add degree symbols in Table S2

Figure S5 – slope lacks units – is this a slope per year (LCC changes on a yearly basis) or per decade? Please clarify and check all physical units carefully in the text and Supp Materials

Figure S6 – several things to fix – first zoom between 6 and 8 on the y-axis (LCC) – second, you need to comment on the range of values – historical GCM data (even calibrated) should not be able to reproduce the observed interannual variability in temperature (only the trends as the GHG signal is included). They are coupled climate models initialized at the start of 1900ish

Figure S7 – I liked the comparison. Perhaps the authors could interpolate the final LCC estimates to the same spatial grid (NASA-NEX at 0.5x0.5 degrees) and plot a difference map (as Fig S7c) – this would make the comparison/discussion clearer

For the 200mm threshold – the authors can also mention that 200mm annual rainfall is a standard

threshold to delineate desert regions – according to Koppen-Geiger if I remember
https://en.wikipedia.org/wiki/K%C3%B6ppen_climate_classification

Reviewer #2 (Remarks to the Author):

Thanks for the resubmission. First of all apologies for the multiple references to dengue/arbovirus transmission maps. While I do believe many of the points made in the original review are also valid for *Aedes* population dynamics, I acknowledge that they were imprecisely made. I'd also thank the reviewers for improving their validation dataset, I think this is a useful addition to that part of the work. However, I do still think there are still two areas of contention that still exist, novelty of the work and propagation of uncertainty in model parameters.

Novelty:

I think I would still contest most of the author stated novelties of this analysis:

First application of a mathematical development stage model to global scale

- See Hopp and Foley *Climate Change* 2001 38:441-463. These models have been around for nearly 20 years and over time have become more disease focussed and now more statistical in nature due to a number of driving motivations, but there definitely are global developmental stage models already published. I could not see what the clear modelling advance in the paper was over this older work.

Use of daily climate data

- Yes true, not aware of other studies doing this with the latest daily climate data

Climate change is causing an acceleration in environmental suitability for the development of *Ae. aegypti* by 2050

- Ok, so there's been fairly good evidence that transmission potential for dengue will increase around 2050 due to its effects on *Aedes aegypti* survival, gonotrophic cycle length, and therefore implicitly population development rate, since Patz et al. 1998 *Environ. Health. Pers.* There have also been numerous recent mapping analyses that confirm this using more up-to-date climate projections, more detailed mechanisms and consideration of variables other than temperature (Ryan et al. *PLoS NTDS* 2019, Kraemer et al. *Nature Microbiol.* 2019). Given this, is it a reasonable hypothesis to expect *Ae. aegypti* development potential NOT to increase by 2050?

Changing seasonality might increase number of life cycles and lead to longer peak seasons

- I would argue that the ability of this model to represent seasonal patterns is not tested in this paper (data varies in space but not in time). Sounds feasible, but could also argue that rainfall patterns and their restriction on larval habitat might be a more limiting factor both now and in the future

I am not sure my concern about lack of novelty is going to be resolved with author changes over a reasonable timescale and is therefore maybe more up to the editor. In respect to this, I am just one reviewer and other reviewers' perspectives may differ.

Uncertainty:

Perhaps I could have explained this better. I was referring to specifically the parameters described

between lines 389 and 418. They currently only have one unique value, but should be defined by a distribution given the uncertainty due to measurement error, local adaptation, sub-species differences, etc, etc. How sensitive are your findings to precise measurements of these parameters? The other reason this is important is because currently figures 3 and 5 have no Cis (even though the legend for figure 5 mentions Cis). While the median predictions may fit the validation set well, propagation of uncertainty is also crucial for future projections and would help readers better interpret claims such as changes in seasonality (Figure 5). Given the rigour that goes in to propagating uncertainty in the climate data, it seems reasonable to expect similar from the mosquito development model.

Reviewers' comments:

Reviewer #1 (Remarks to the Author):

Review of "Accelerating invasion potential of disease vector *Aedes aegypti* under climate change"

Manuscript number: NATCOM-19-16916A

Authors: Iwamura et al., 2019

Recommendation: Major revisions - still

The paper greatly improved since the last round of revisions.

Many thanks - Please see below responses to this round of comments.

However, I still spotted important points that deserve clarifications. The manuscript plan needs to be restructured a bit as well (validation section should come first in results); and there are minor points to address before the paper can be accepted for publication in Nature communications.

Major points:

A - Model validation should come first in the results section. The authors also need to state that they carried out a two level validation – one at global scale (using points in and out of their suitability surface) – one more detailed for Mexico (this needs to be spelled out in the text)

We have now moved the validation section to come first in the Results section, and have clarified the two validation steps conducted: “We conducted model validation at two levels – one at the global scale from the occurrence point dataset and another at a local scale using a mosquito abundance dataset.” (L. 96-97)

B - Items in Supplementary Materials should be cited in a chronological order in the text (starting with Table S1 – Table S2 etc - Fig S1 - S2 etc) so the authors will have to reshuffle the items in Supp Material accordingly.

Done.

C – Something important I missed in the former round of revisions (sorry about that) – the authors state that Invasion fronts in North America and China are projected to accelerate from ~2 to 6km/yr by 2050. The recent work by Kraemer et al 2019 [Fig 1 at <https://www.nature.com/articles/s41564-019-0376-y>] shows that historical invasion fronts (derived from observed vector presence data from 1990 to 2010) were more likely in the 60 km/year range for the US and 100km/year for Europe. Thus, there is a large discrepancy between simulated invasion fronts for the future based on this study (about 20ish km in 10 years) and recently observed invasion fronts (about 60 or 100 km in one year for temperate regions – and this is consistent with the observed spread of *Ae. albopictus* from Italy to the north of France in roughly 10ish years). This might be related to the fact that the mosquito already invaded its

potential niche during the recent context and it will slowly spread in future due to climate change (and this finding is consistent with the Kraemer paper). Other recent studies on a similar topic should also be discussed further in discussion (Monaghan work- <https://journals.plos.org/ploscompbiol/article?id=10.1371/journal.pcbi.1007369> - Kraemer work on invasion front <https://www.nature.com/articles/s41564-019-0376-y> etc). The authors need to discuss this point further in discussion (perhaps use relative changes and state that the historical invasion fronts estimated from their model are quite small with respect to rates derived from observations for Europe and the USA).

We appreciate these suggestions, and note the importance of referencing these studies. However, Kraemer et al. 2019 did not calculate the speed of ‘invasion frontiers’ in the same way we did. Instead, they calculated ‘localised invasion velocity’ for ‘short-range importation between adjacent districts’. In contrast, we estimate the speed at which landscapes are predicted to become ‘suitable’ for *A. aegypti* beyond their current range edges. We have now clarified this distinction with reference to the additional role of importations in *A. aegypti* range expansions as “These patterns imply that sudden shifts in invasion frontiers should be expected as changing underlying suitability interacts non-linearly with human introduction and dispersal processes (but see 39 for localised speed of the importation of *A. aegypti* within established species range).” (L. 261-264)

NOTE: We have restricted our revisions to those relevant to *A. aegypti* as we did not study *A. albopictus* here (according to Kraemer et al. 2019, the localised invasion velocity of *Ae. aegypti* is 250 km per year, much faster than *Ae. Albopictus*).

Minor comments:

L19-20: “change in human disease risk” - singular sounds better

Done

L20-21: "we develop a mechanistic phenology model..." - you can state the novelty of your study there: "we develop a mechanistic phenology model derived from agriculture and applied it to..."

Done – We added as “We develop a mechanistic phenology model derived from the study of agricultural pests and apply it to *Aedes aegypti*, an invasive mosquito vector for impactful arboviruses (e.g. dengue, zika).” (L. 20-22)

L21: "an invasive vector of impactful arboviruses"

Done

L26: "while this trend will accelerate..." - when you mention the future and risk simulations, everything is hypothetical (so avoid deterministic statement like this) so reword to "while this trend is simulated to..."

Done. We applied similar edits throughout the ms to address this comment.

L28: "during peak seasons" - "during warm seasons". Last sentence of the abstract: "An increase in LCC combined with a lengthening of the transmission season is simulated to accelerate ..."

Done

L41: "...are often linked to or partially predicted by the distribution..." - "are often conditioned by"
Note this is not a necessary true conditions

Done

Theoretically, climate conditions might be suitable for a particular vector (allowing development and survival), but temperature conditions might still be too low to enable the insect vector to become infectious - in other words they can survive but won't be able to transmit disease in location X or Y. As an example, some endemic UK mosquito species can transmit exotic pathogens (like Zika virus for example) in a lab setting at high temperatures (work in progress in our team), but they won't be able to transmit the pathogen in the field (so temperature are still too cold to allow mosquitoes to become infectious). I could have made that comment earlier but this needs to be discussed in the manuscript.

We agree entirely. As suggested, we have added the following in the Discussion (L305-310, in the paragraph of further research to improve our model): "It is also important to note here that while climate conditions might be suitable for the development and survival of a vector, the conditions for effective disease transmissions may be different. Integrating a disease transmission component into our models or, conversely, integrating our phenology model into existing disease transmission models could help bridge the gap between predicting global change impacts on vectors versus the realized health impacts."

L50-51: "to assess the static or changing suitability of landscapes or regions for the species" - "to assess the stability (or stationarity) of landscapes or regions for the species"

We edited as "to assess the suitability of".

L70 - (Linnaeus, 1762) - ok but remove it - no need to cite the father of taxonomy in here I would say

We have kept this citation to be aligned with other articles in the field (citing naming authority for the species, which in this case is also the same person as the 'father of taxonomy').

L71 - "for predicting invasive pest establishments..." - "for predicting invasive pest establishments"

affecting agriculture production" - mention agriculture in that sentence and you can say these methods have not been extensively applied to medical entomology (to state the novelty of your study further).

Done

L84: "...key development rates in mosquitoes..." - "development rates of mosquitoes at different stages"

Done

L85: RCP8.5 (remove space between RCP and scenario ID - double check the whole doc as I spotted this several time).

Done

"RCP4.5 and RCP 8.5 climate change scenarios, which reflect differences in the degree to which greenhouse gas emissions and consequent climatic changes may be curbed by the middle of this century". Ref 32 refers to a VBD modelling paper - not the official RCPs- authors should cite the standard Moss et al 2010 <https://www.nature.com/articles/nature08823> or Van Vuren et al 2011 paper <https://link.springer.com/article/10.1007/s10584-011-0148-z>

Done

Fig 1 caption - "to the threshold used in Fig 4" - prefer "to the threshold used in subsequent analysis"

Done

Fig 3 – typo in legend - "equatorial" should be "equatorial"

Done

Photoperiod is important for *Ae. albopictus* and it might be a limiting factor at high latitudes (mention this somewhere in discussion)

Done – although we deal with *Ae. aegypti* (not *albopictus*), we now mention photoperiod alongside other covariates: “or mechanistic/process-based models, which make use of a species’ physiological responses to specific environmental parameters (e.g., temperature, rainfall, humidity, photoperiod; see ¹³)” (L.47-48).

L160 - Generic comment do not say: [see Discussion] in the text - discuss your results then mention important points missing in discussion please

Done

Fig 5 caption - when referring to geographical domain use standard degrees north / south system eg [40S to +40°N for example] - or the reader might be confused with degrees Celsius (for temperature)

Done

L219: "from something to something".

Done

L229: "reported for Mexico by" - you need to state that you have extra validation analysis for Mexico somewhere in Methods.

Done. See above for our response to the earlier validation comment.

L346: "eggs, larvae, pupae and emergence" - "eggs, larvae, pupae and adults"

Done

Figure S1 - I am unsure I think some arrows are missing on Fig S1 (connecting eggs to life cycle)

Agree. Figure has now been corrected.

Table S1: The authors should highlight AUC above 0.7 (in bold – good scores) – AUC below or equal to 0.5 threshold are not better than random – so use a color or star code to discriminate good and poor AUC values at country scale. The low AUC values for Brazil are not surprising (as models tends to forecast the whole country as suitable – and surveillance data is mostly available for the populated districts / eastern regions), and I assume that collecting mosquitoes in the Amazon is quite complicated.

Done – but we avoided color coding as AUC scores are typically more conservatively interpreted in indicative rather than concrete bands (colour coding could give the impression that the rule-of-thumb bands (good, excellent etc) have a more robust statistical basis for description). In the Results section (L111-113), we similarly hypothesise on the reasons for Brazil’s low AUC value: “Brazil shows low AUC (0.35) likely due to the expanse of the Amazon basin which has a highly suitable climate for *A. aegypti* but very low human presence.”

Table S2: Add degree symbols

Done

Figure S5 – slope lacks units – is this a slope per year (LCC changes on a yearly basis) or per decade? Please clarify and check all physical units carefully in the text and Supp Materials

Here, slope means change in LCC per month, reflecting that the statistic was calculated from underlying monthly time series data. We have now indicated this in the figure caption.

Figure S6 – several things to fix – first zoom between 6 and 8 on the y-axis (LCC) – second, you need to comment on the range of values – historical GCM data (even calibrated) should not be able to reproduce the observed interannual variability in temperature (only the trends as the GHG signal is included). They are coupled climate models initialized at the start of 1900ish

Done – We fixed the figure and now the discrepancy between ERA 5 and GDDP is discussed as “While the outputs are not equivalent as expected (GDDP is based on GCMs, and ERA 5 is calculated from observation), they were strongly correlated between datasets ($r^2 = 0.53$, $p < 0.001$).” (SI Figure S7).

Figure S7 – I liked the comparison. Perhaps the authors could interpolate the final LCC estimates to the same spatial grid (NASA-NEX at 0.5x0.5 degrees) and plot a difference map (as Fig S7c) – this would make the comparison/discussion clearer

The ERA5 and NASA-NEX have comparable resolution (ERA5 is roughly at 30 by 30 km, while NASA-NEX is at roughly 25 by 25km). We followed this suggestion to create a difference map, which is now included as Figure S7c.

For the 200mm threshold – the authors can also mention that 200mm annual rainfall is a standard threshold to delineate desert regions – according to Köppen-Geiger if I remember https://en.wikipedia.org/wiki/K%C3%B6ppen_climate_classification

Thank you. This has been already included as in L425-427 (Methods Section): “Based on the Köppen-Geiger climate classification⁷² for dry regions and the literature^{73,74}, the precipitation threshold was set to less than 200 millimetres of annual rainfall, which contains 99% of observation records.”.

Reviewer #2 (Remarks to the Author):

Thanks for the resubmission. First of all apologies for the multiple references to dengue/arbovirus transmission maps. While I do believe many of the points made in the original review are also valid for Aedes population dynamics, I acknowledge that they were imprecisely made.

No problem, thanks.

I'd also thank the reviewers for improving their validation dataset, I think this is a useful addition to that part of the work.

Thank you.

However, I do still think there are still two areas of contention that still exist, novelty of the work and propagation of uncertainty in model parameters.

Please see below for detailed responses to these concerns.

Novelty:

I think I would still contest most of the author stated novelties of this analysis:

First application of a mathematical development stage model to global scale - See Hopp and Foley Climate Change 2001 38:441-463. These models have been around for nearly 20 years and over time have become more disease focussed and now more statistical in nature due to a number of driving motivations, but there definitely are global developmental stage models already published. I could not see what the clear modelling advance in the paper was over this older work.

Hopp and Foley 2001, and the earlier CIMSiM model studies on which this study was built, is an excellent contribution to the field, and we now mention its importance and relevance to our study in the introduction. However, we disagree that this and other related/subsequently developed studies undermine the novelty that we provide in our study, although we recognize the need to be even clearer with our novelty statements.

For example, while Hopp and Foley 2001 modified an existing and widely used developmental stage model (CIMSiM) to examine “the development, population dynamics, and potential distribution of the *Aedes aegypti*” in a climate change context and with gridded climate data, their methods and applications are fundamentally different from ours. Most notably, they discount the use of growing degree day (GDD) models for development, which forms the foundation of our phenology model, citing “Many entomological models use day-degree or temperature summation models that assume development rate is proportional to temperature. This suits modeling development within a limited temperature range”, before going on to use a completely different kind of development rate model (enzyme kinetics).

We counter that our GDD model is by no means limited in the way described by Hopp and Foley (2001) above. Rather, when applied to estimate development rates (not done in our study, since ours is a mechanistic start-stop model that does not directly use rate estimations to calculate LCC), we are able to recreate biologically reasonable ‘temperature response curves’ for each life stage that refute the suggestion that they are simply ‘proportional to temperature’. As such, and given the relative simplicity and completely different applications we make, we are

confident that our contribution provides a high degree of novelty that will be appreciated by readers across diverse fields.

Use of daily climate data

- Yes true, not aware of other studies doing this with the latest daily climate data

Thanks. Importantly, by using the daily dataset, we can maximize the benefits of the GDD model by analyzing the impacts of varying temperature to developmental speed. The LCC outputs are the key indices for invasion analyses and seasonal analyses, so it is difficult to separate this dataset and novelty of the research.

Climate change is causing an acceleration in environmental suitability for the development of *Ae. aegypti* by 2050

- Ok, so there's been fairly good evidence that transmission potential for dengue will increase around 2050 due to its effects on *Aedes aegypti* survival, gonotrophic cycle length, and therefore implicitly population development rate, since Patz et al. 1998 Environ. Health. Pers. There have also been numerous recent mapping analyses that confirm this using more up-to-date climate projections, more detailed mechanisms and consideration of variables other than temperature (Ryan et al. PLoS NTDS 2019, Kraemer et al. Nature Microbiol. 2019). Given this, is it a reasonable hypothesis to expect *Ae. aegypti* development potential NOT to increase by 2050?

We may differ fundamentally in our views here. Our results are based on rigorous study and show that environmental suitability increases for *Aedes aegypti* are also accelerating. As the reviewer said, we are not aware of any other studies that have explicitly tested for or predicted this. In other climate change studies, accelerating trends have received a lot of recent attention (e.g., accelerating rate of sea level rise in a recent Nature Climate Change paper - <https://www.nature.com/articles/s41558-019-0531-8>). Accelerating suitability has a number of additional implications than simply increasing suitability, and we think this is one of the more novel and important results from our study that will be widely appreciated by readers.

Changing seasonality might increase number of life cycles and lead to longer peak seasons

- I would argue that the ability of this model to represent seasonal patterns is not tested in this paper (data varies in space but not in time). Sounds feasible, but could also argue that rainfall patterns and their restriction on larval habitat might be a more limiting factor both now and in the future

We agree that further research is required on seasonality, but we counter again that this does not undermine our contribution to the extent that it is not worth reporting as a novel result. We do also in fact show the LCC is well correlated with monthly abundance counts in the local study in Mexico that we employed for one of our validation tests, encouraging us to explore and report the changing basis of seasonality in our models. More broadly, there are in fact very few studies that are taking aim at predicting the changing basis of seasonality of disease vectors under

climate change, despite the fact that this is one of the most commonly cited mechanisms that researchers use to link climate change to changes in infectious disease risks. At the very least, our model provides some testable predictions that others may find useful or stimulating.

I am not sure my concern about lack of novelty is going to be resolved with author changes over a reasonable timescale and is therefore maybe more up to the editor. In respect to this, I am just one reviewer and other reviewers' perspectives may differ.

We very much appreciate the depth of consideration and ideas shared in the reviews, which have substantially tightened and improved our contribution. We very much hope that our replies above and the revisions to the text we have made have more firmly established the novel elements of our study and, more importantly, the novel combination of multiple elements in our study to make progress, stimulate interest and management considerations on this important and dynamic topic.

Uncertainty:

Perhaps I could have explained this better. I was referring to specifically the parameters described between lines 389 and 418. They currently only have one unique value, but should be defined by a distribution given the uncertainty due to measurement error, local adaptation, sub-species differences, etc, etc. How sensitive are your findings to precise measurements of these parameters? The other reason this is important is because currently figures 3 and 5 have no Cis (even though the legend for figure 5 mentions Cis). While the median predictions may fit the validation set well, propagation of uncertainty is also crucial for future projections and would help readers better interpret claims such as changes in seasonality (Figure 5). Given the rigour that goes in to propagating uncertainty in the climate data, it seems reasonable to expect similar from the mosquito development model.

We agree that uncertainty in the simulation must be addressed. This really involves two different issues – a) sensitivity analyses on model parameters and b) presenting CIs for predictions to capture the uncertainty related to the use of multiple climate models.

a) Sensitivity analyses

In this revision, in addition to those previously included relating to rainfall threshold (200 vs 900 mm/yr) and the differences in climate data sets (observation vs simulated), we have now additionally included sensitivity analyses for a range of relevant parameters in the phenology model, namely oviposition (incl. blood feeding), cold-kill and heat-kill conditions (L.432-437).

A summary of the results, now documented in the SI, follows:

- 1. Blood feeding and oviposition: We ran models with 75%, 125% and 150% developmental speed compared to the default growth speed curve (which is based on integrated literature review). In the 75% scenario, the days required**

for oviposition at a certain temperature takes 75% of the default length (e.g. instead of 10 days at the temperature 30°C, it takes 7.5 days). In the 125% scenario, everything takes 25% longer (e.g. instead of 10 days, it takes 12.5 days).

→ While the days necessary for oviposition has altered values of LCC for 125% (higher than the default value), LCC became less sensitive to this parameter at even higher values (150%). Despite changes in baseline figures, the overall trends are very similar to default setting. See SI Fig S6a.

2. **Cold-kill conditions:** Based on the literature, we previously set the cold kill temperature as 0°C (with 3 months of the cold-kill temperature, they cannot colonize). We tested three other conditions: -5°C, 5°C, 10°C.

→ The results are overall robust to the choice of cold-kill conditions. Lowering the threshold to -5°C and increasing it to +5°C produced similar results with the default (0°C), with +5°C affecting LCC more. The decreasing in the LCC is more apparent in +10°C setting, but the trend one again remains the same. (SI Figure S6b).

3. **Heat-kill condition:** Based on the literature, we previously set the heat kill condition as 38°C. We ran models to explore the effect of using 36°C and 40°C.

→ The results are highly robust to the choice of heat-kill condition. The results are nearly identical until around 2020. Then slight differences between the settings are observed further into the future (when climate become warmer), indicating this may also affect LCC slightly at warmer climates. (SI Figure S6c). Once again, however, the overall trends and main implications and conclusion remain the same as those reported for the default value.

b) Confidence Intervals

Uncertainty around the LCC predictions related to differences in climate models was already presented in the text where relevant, and was also provided in Figure 5 (seasonality plots) as well as for the trend estimations (Fig S4, slope/rate of change estimates).

With respect to Fig. 3, we feel that the figure is too considerably degraded (too much overplotting) in its interpretability by the inclusion of CI bands when we present the mean trends broken down by RCPs for climate regions (Fig 3a) or continental regions (Fig 3b). We have thus modified the captions to make mention of this purposeful omission, and point readers to the text and Fig 5, where a measure of the magnitude of this uncertainty related to GCM differences is clearly provided.

REVIEWERS' COMMENTS:

Reviewer #1 (Remarks to the Author):

Review of "Accelerating invasion potential of disease vector *Aedes aegypti* under climate change"

Manuscript number: NATCOM-19-16916B

Authors: Iwamura et al., 2019

Recommendation: Minor revisions

The paper greatly improved since the last round of revisions. I just have minor comments

Minor comments:

L18 - "while climate change is exacerbating their risk"

Risk might also decrease in some warm regions (when temperatures will exceed survival thresholds for mosquito insects – perhaps 2080s in the warmest regions, eg the Sahel, central Australia, the Middle East etc). Thus prefer - "climate change is expected to impact their risk"

L20: "in human disease risk".

L22: "dengue, zika and Yellow Fever".

L50: "to assess changing suitability of landscapes...".

L71: "for predicting invasive pest establishments affecting plant health" or "in agriculture".

L84: "backcasts" - I am used to "hindcast" (used in seasonal forecasting) or perhaps used "historical projections" as you are using standard calibrated historical GCM simulations from the IPCC.

L98: "for further details"

"increased through time from a 5-year average of 7.08" - "increased from 7.08 ... to..." No need to specify 5y averages - this is already captured by "(1950-1954 average)"

L102-104: ". Projections to the 2050s suggest this trend will accelerate, with the 5-year (2050 - 2054) average number of generations per year predicted to increase by a further 17.1% (12.4-21.8%) under RCP 4.5 and 24.3% (18.5-30.0%) under RCP 8.5."

Perhaps: "Future projections suggest this trend will accelerate, with the average number of generations per year predicted to increase by a further 17.1% (12.4-21.8%) by the 2050s under RCP4.5 and 24.3% (18.5-30.0%) under RCP8.5." – Define key time periods (e.g. 2050s as the 2050-2054 average and do the same for other time slices) in Methods - then use 2050s etc throughout the text.

L109: "Equivalent estimations using 10-year averages returned similar results" - "Estimations using 10-year averages yielded similar results"

L111-L124: "Figure 2 illustrates these changes relative to the midpoint of the time series (5-year average 2000-2004), with increases in up to 6 LCC per year in tropical areas observed since 1950 (1950-1954 average) and a further 6-10 LCC per year expected by 2050 (2050-2054 average) in some areas, with the greatest increases predicted under RCP8.5. Similar increases are observed using a 10-year average (SI Table S4)"

Long sentence - reword - perhaps:

"Figure 2 illustrates LCC changes with respect to the 2000-2004 average. LCC increased up to 6 LCC per year in tropical areas since the 1950s (1950-1954 average) and a further 6-10 LCC per year expected by the 2050s (2050-2054 average) in some areas. The greatest increase is predicted under RCP8.5. It is noteworthy that similar increases are simulated using 10-year averages (SI Table S4)"

L123-124: "The increase in mean and rates of simulated LCC differs significantly across geographic and climatic regions"

L126-27: "while in Europe, North America, West and Central Asia"

L132: "under RCP4.5 and RCP8.5" – no space between RCP and 4.5 or 8.5 – double check in the text and Supp Materials, as I found this mistake several times.

L133: "more moderate increases in LCC observed historically" - prefer "simulated during the historical period". You are using simulation of LCC estimates so be very careful with the semantic throughout the text (observation might refer to climate observation or disease / vector observation) – mention "simulated" or "simulations" when you refer to LCC.

L136-139: "For example, the rate of change in LCC per year at 0-10S in the period 2000-2050, as indicated by the Sen slope indicator, is projected to increase 2.5- and 3.9-fold relative to the historical increase (1950-2000) for RCPs 4.5 and 8.5, respectively (see SI, Fig. S5)."

This sentence is too long and poorly structured - please reword using two sentences.

L157: "so this was set as the threshold" - the 2.5% threshold should be mentioned in Methods with some explanations about why - then there is no need to repeat the information in the results section (or do it very briefly).

First paragraph of "Invasion frontiers" - the statements are quite vague in some places (increase / decrease) and lack geographical details. Be a bit more descriptive about new regions at risk (eg describe the new geographical regions at risk or not - "over the Mediterranean basin etc" Results discussion looks ok for China

The results for Europe look very conservative eg the future risk maps look like past historical circulation of DENV over Southern Europe but you now mention this caveat in discussion so it's fine

Results for China looks good - you can highlight the hotspots in Guangzhou and Guangdong - where DENV has circulated heavily over the region. There is a huge amount of publications on dengue epidemics in Guangzhou and risk models to cite and discuss further in discussion

Paragraph starting L184 needs to be improved

Use "mean seasonal cycle of simulated LCC" and be more descriptive about LCC seasonality changes simulated during the standard boreal seasons (winter - spring - summer) per climatic regions.

L200-201: "The middle latitudes (10-20 and 20-30°) will be most affected with seasonal changes in LCC, LCC increase is more significant under the RCP8.5 scenario".

L205: "when a LCC > 10 is set as a threshold." - I spotted several minor glitches with the English overall - the manuscript will benefit from a read/good scan by a native English speaker

Another example: "which has a highly suitable climate for *A. aegypti* but very low human presence" - "*Ae. aegypti* but with very low human population densities"

"reported by Lozano-Fuentes et al. (2012) and Moreno-Madriñán et al. (2014)" - ok but you need to say where and when - I assume the observation data used for validation was gathered in Mexico over the period year1-year2?

L256-> "The seasonal trend analysis further indicates shifting patterns in the seasonality of mosquito life-cycle completions, whereby both favourable periods have extended and increases in peak LCC are observed in both backcasts and forecasts, reflecting recent and anticipated climate change."

This sentence is confusing and too long - please reword - see also former comment on backcast (use historical simulations preferably or hindcast (hindcast could be misleading too as it is commonly employed in seasonal forecasting studies)

L269 paragraph - ok you mention that your future model projections appear conservative with respect to the historical background in discussion so it's fine

L324: "*Ae. aegypti* is an ideal species "

L444: "the period 1980-2005" - typo - "the period..."

L3: "GDD is calculated based the model's temperature input using the formula: " - "GDD is calculated based on temperature input using the formula: "

In Supp Materials and the text (several occurrences) - "*A. aegypti*" should be "*Ae. aegypti*" - *Ae.* for *Aedes* and use italics for species name; no space between RCP and number (RCP4.5 or RCP8.5 etc)

Table S1: "We did not use Kraemer et al., background dataset based on presence on other mosquito species as in the main text for country level validation because many of the countries with high *A. aegypti* observations often lack observations for other species, leading to AUC inflation" - I struggled to understand what you meant here - please reword & clarify.

REVIEWERS' COMMENTS:

Reviewer #1 (Remarks to the Author):

Review of "Accelerating invasion potential of disease vector *Aedes aegypti* under climate change"

Manuscript number: NATCOM-19-16916B

Authors: Iwamura et al., 2019

Recommendation: Minor revisions

The paper greatly improved since the last round of revisions. I just have minor comments

Thank you.

Minor comments:

L18 - "while climate change is exacerbating their risk"

Risk might also decrease in some warm regions (when temperatures will exceed survival thresholds for mosquito insects – perhaps 2080s in the warmest regions, eg the Sahel, central Australia, the Middle East etc). Thus prefer - "climate change is expected to impact their risk"

Done.

L20: "in human disease risk".

This is no longer relevant as we modified this sentence to respond the requests from the editorial office.

L22: "dengue, zika and Yellow Fever".

Done.

L50: "to assess changing suitability of landscapes...".

Done.

L71: "for predicting invasive pest establishments affecting plant health" or "in agriculture".

Done.

L84: "backcasts" - I am used to "hindcast" (used in seasonal forecasting) or perhaps used "historical projections" as you are using standard calibrated historical GCM simulations from the IPCC.

We have corrected throughout to use 'historical projections'.

L98: "for further details"

Done.

"increased through time from a 5-year average of 7.08" - "increased from 7.08 ... to..." No need to specify 5y averages - this is already captured by "(1950-1954 average)"

Done.

L102-104: "Projections to the 2050s suggest this trend will accelerate, with the 5-year (2050 - 2054) average number of generations per year predicted to increase by a further 17.1% (12.4-21.8%) under RCP 4.5 and 24.3% (18.5-30.0%) under RCP 8.5." Perhaps: "Future projections suggest this trend will accelerate, with the average number of generations per year predicted to increase by a further 17.1% (12.4-21.8%) by the 2050s under RCP4.5 and 24.3% (18.5-30.0%) under RCP8.5." – Define key time periods (e.g. 2050s as the 2050-2054 average and do the same for other time slices) in Methods - then use 2050s etc throughout the text.

Adopted. Now defined in the Methods section as “Throughout the analysis, we used 5-year averages to define and better assess key time periods: we refer to the 1950s as the 1950 – 1954 average, the 2000s as the 2000 – 2004 average, and the 2050s as the 2050 – 2054 average.” (L.479).

L109: "Equivalent estimations using 10-year averages returned similar results" - "Estimations using 10-year averages yielded similar results"

Done.

L111-L124: "Figure 2 illustrates these changes relative to the midpoint of the time series (5-year average 2000-2004), with increases in up to 6 LCC per year in tropical areas observed since 1950 (1950-1954 average) and a further 6-10 LCC per year expected by 2050 (2050-2054 average) in some areas, with the greatest increases predicted under RCP8.5. Similar increases are observed using a 10-year average (SI Table S4)"

Long sentence - reword - perhaps:

"Figure 2 illustrates LCC changes with respect to the 2000-2004 average. LCC increased up to 6 LCC per year in tropical areas since the 1950s (1950-1954 average) and a further 6-10 LCC per year expected by the 2050s (2050-2054 average) in some areas. The greatest increase is predicted under RCP8.5. It is noteworthy that similar increases are simulated using 10-year averages (SI Table S4)"

Adopted. Now reads “Figure 2 illustrates LCC changes with respect to the 2000s average. LCC increased up to 6 LCC per year in tropical areas since the 1950s and a further 6-10 LCC per year is expected by the 2050s in some areas. The greatest increase is predicted under RCP 8.5. Similar increases are estimated using 10-year averages (Supplementary Table 4).” (L. 142)

L123-124: "The increase in mean and rates of simulated LCC differs significantly across geographic and climatic regions"

Adopted. Revised to “The overall suitability, the increase in mean and rates of simulated LCC differ significantly across geographic and climatic regions”.

L126-27: "while in Europe, North America, West and Central Asia"

Done.

L132: "under RCP4.5 and RCP8.5" – no space between RCP and 4.5 or 8.5 – double check in the text and Supp Materials, as I found this mistake several times.

Done.

L133: "more moderate increases in LCC observed historically" - prefer "simulated during the historical period". You are using simulation of LCC estimates so be very careful with the semantic throughout the text (observation might refer to climate observation or disease / vector observation) – mention "simulated" or "simulations" when you refer to LCC.

Agreed in general to emphasize that our results are simulated, as we had corrected in the previous revision. But in this case it is misleading because we did not simulate 'moderate increase' – such trends are found in the model results. Here we added a distinction "in our results during the historical period."

L136-139: "For example, the rate of change in LCC per year at 0-10S in the period 2000-2050, as indicated by the Sen slope indicator, is projected to increase 2.5- and 3.9-fold relative to the historical increase (1950-2000) for RCPs 4.5 and 8.5, respectively (see SI, Fig. S5)." This sentence is too long and poorly structured - please reword using two sentences.

We do not wish to split this sentence into two as this impacts readability.

L157: "so this was set as the threshold" - the 2.5% threshold should be mentioned in Methods with some explanations about why - then there is no need to repeat the information in the results section (or do it very briefly).

Adopted. Explanation now appears in the Methods: "To do this, we defined an invasion frontier contour as a contour line representing the LCC value below which 2.5% of *Ae. aegypti* occurrence records globally occurred, representing uncommon but demonstrated establishment at the lower end of the LCC distribution. Globally, this 2.5% contour line corresponds with the areas ≤ 10 LCC per year, so this was set as the invasion frontier threshold. We then tracked this invasion frontier contour through time to illustrate which new areas could become suitable for future establishment and by when." (L. 487).

The result now reads "Contour lines indicating invasion frontiers (≥ 10 LCC; see Methods) were used to examine expansion in suitable areas in the three focal regions over multiple periods (Fig. 4)." (L. 165)

First paragraph of "Invasion frontiers" - the statements are quite vague in some places (increase / decrease) and lack geographical details. Be a bit more descriptive about new regions at risk (eg

describe the new geographical regions at risk or not - "over the Mediterranean basin etc". Results discussion looks ok for China. The results for Europe look very conservative eg the future risk maps look like past historical circulation of DENV over Southern Europe but you now mention this caveat in discussion so it's fine.

Results for China looks good - you can highlight the hotspots in Guangzhou and Guangdong - where DENV has circulated heavily over the region. There is a huge amount of publications on dengue epidemics in Guangzhou and risk models to cite and discuss further in discussion.

Thank you very much for pointing out the case in China. We now mention this case from China in the Results and Discussion sections. We cite the most recent outbreak analysis (Zhu, G., Xiao, J., Liu, T., Zhang, B., Hao, Y., & Ma, W. (2019). Spatiotemporal analysis of the dengue outbreak in Guangdong Province, China. *BMC infectious diseases*, 19(1), 493. <https://doi.org/10.1186/s12879-019-4015-2>)

Results section now reads: "In the USA, the model suggests that the south-eastern states (i.e. Florida, Arizona, Texas) have already seen the advancement of an invasion frontier, as is also supported by observations of *Ae. aegypti* occurrence expanding there. The model confirms relatively slow invasion frontier expansion in China, but predicts more rapid advancement under future climates, including in recent dengue outbreak hotspots (Guangzhou and Guangdong provinces)³⁷. In Europe, this suitability threshold is patchier, restricted to the southern margins historically, yet clearly increasing suitability in other places (e.g., over the Mediterranean basin) in the future. Continuous stretches of suitability across Europe are not observed even under RCP 8.5 by 2050 (Fig. 4)." (L. 166)

Discussion section now reads: "The model also sheds light on the idiosyncrasies among regions in the way changing environmental conditions will facilitate vector invasion. Our results predict that invasion frontiers, representing expanding regions that are environmentally suitable for this species, in China and USA are predicted to advance 2.4-3.5 times faster by 2050 (5.2-6.0 km yr⁻¹) than was estimated through historical projections (1950-2000). Europe is expected to experience isolated areas of sustained suitability for *Ae. aegypti* in Spain, Portugal, Greece and Turkey by 2030. In China, our model predicts expansion of frontiers into the Guangzhou and Guangdong provinces, where dengue outbreaks have been reported recently³⁹. These patterns imply that sudden shifts in invasion frontiers should be expected as changing underlying suitability interacts non-linearly with human introduction and dispersal processes (but see ⁴⁰ for localised speed of the importation of *Ae. aegypti* within established species range)." (L. 223)

Paragraph starting L184 needs to be improved. Use "mean seasonal cycle of simulated LCC" and be more descriptive about LCC seasonality changes simulated during the standard boreal seasons (winter - spring - summer) per climatic regions.

The seasonal changes at different climate regions are described in the next paragraph. Considering limited space, we do not feel such repetition is necessary. We modified "seasonal profile of LCCs" to "seasonality profiles of simulated LCCs".

L200-201: "The middle latitudes (10-20 and 20-30°) will be most affected with seasonal changes in LCC, LCC increase is more significant under the RCP8.5 scenario".

Done.

L205: "when a LCC > 10 is set as a threshold." - I spotted several minor glitches with the English overall - the manuscript will benefit from a read/good scan by a native English speaker.

Done.

Another example: "which has a highly suitable climate for *A. aegypti* but very low human presence" - "*Ae. aegypti* but with very low human population densities"

Done.

"reported by Lozano-Fuentes et al. (2012) and Moreno-Madriñán et al. (2014)" - ok but you need to say where and when - I assume the observation data used for validation was gathered in Mexico over the period year1-year2?

This information is already in the methods: "Second, we compared LCC predictions with *Ae. aegypti* abundance data from ⁸⁴, who conducted *Ae. aegypti* abundance surveys in villages distributed across a large elevational gradient (0 – 2000m) in central Mexico^{84,85}. This study was conducted in 2011 and represents the best available case study that we are aware of focussing on *Ae. aegypti* abundance. The surveys spanned a relatively large geographic area (300x100km) and, critically, employed the same sampling methodology across all surveys. We utilised data from all villages in the study, excluding Orizaba and its 'dormitory' city of Rio Blanco⁸⁵, leaving surveys from 10 villages for analysis (see Supplementary Table 2). For the validation analysis, we extracted the average LCC of our model at the locations for these villages in the same year of the sampling effort (2011) and correlated these values against abundance estimates from the surveys (adult mosquitos). As such our comparison is relative rather than absolute." (L.459). The details for the sampling sites are listed in the Supplementary Table 2.

L256-> "The seasonal trend analysis further indicates shifting patterns in the seasonality of mosquito life-cycle completions, whereby both favourable periods have extended and increases in peak LCC are observed in both backcasts and forecasts, reflecting recent and anticipated climate change."

This sentence is confusing and too long - please reword - see also former comment on backcst (use historical simulations preferably or hindcst (hindcast could be misleading too as it is commonly employed in seasonal forecasting studies).

Adopted. This now reads: "The seasonal trend analysis further indicates shifting patterns under changing climates in the seasonality of mosquito life-cycle completions, whereby both longer periods of favourable conditions and higher intensity in peak LCC are observed in both historical and future projections." (L. 234).

L269 paragraph - ok you mention that your future model projections appear conservative with respect to the historical background in discussion so it's fine.

Thank you.

L324: "Ae. aegypti is an ideal species "

Done.

L444: "the peirod 1980-2005" - typo - "the period..."

Done.

L3: "GDD is calculated based the model's temperature input using the formula: " - "GDD is calculated based on temperature input using the formula.

Done.

In Supp Materials and the text (several occurrences) - "A. aegypti" should be "Ae. aegypti" –Ae. for Aedes and use italics for species name; no space between RCP and number (RCP4.5 or RCP8.5 etc)

Adopted throughout.

Table S1: "We did not use Kraemer et al., background dataset based on presence on other mosquito species as in the main text for country level validation because many of the countries with high A. aegypti observations often lack observations for other species, leading to AUC inflation" - I struggled to understand what you meant here - please reword & clarify.

We clarified: "For country-level validation, we did not use Kraemer et al., background dataset which contains the observations on the presence of other mosquito species because many of the countries with high *Ae. aegypti* observations often lack observations for other species, leading to AUC inflation." (Supplementary Table 1)